# Breakthrough Sensor-Limited Single View: Towards Implicit Temporal Dynamics for Time Series Domain Adaptation

**Mingyang Liu[1], Xinyang Chen[1✉], Xiucheng Li[1], Weili Guan[2], Liqiang Nie[1]**

[1]School of Computer Science and Technology, Harbin Institute of Technology (Shenzhen)

[2]School of Information Science and Technology, Harbin Institute of Technology (Shenzhen)

{mingyangliu1024, chenxinyang95, nieliqiang}@gmail.com, {lixiucheng,guanweili}@hit.edu.cn

## Abstract

Unsupervised domain adaptation has emerged as a pivotal paradigm for mitigating distribution shifts in time series analysis. The fundamental challenge in time series domain adaptation arises from the entanglement of domain shifts and intricate temporal patterns. Crucially, the latent continuous-time dynamics, which are often inaccessible due to sensor constraints, are only partially observable through discrete time series from an explicit sensor-limited single view. This partial observability hinders the modeling of intricate temporal patterns, impeding domain invariant representation learning. To mitigate the limitation, we propose **EDEN** (multiple **E**xplicit **D**omain **E**nhanced adaptation **N**etwork), expanding the raw dataset to multi-scale explicit domains, multi-subspace explicit domains and multi-segment explicit domains. EDEN enhances domain adaptation with three coordinated modules tailored to integrate multiple explicit domains: (1) Multi-Scale Curriculum Adaptation implements progressive domain alignment from coarse-scale to fine-scale. (2) Quality-Aware Feature Fusion evaluates feature quality in multi-subspace explicit domains and adaptively integrates temporal-frequency features. (3) Temporal Coherence Learning enforces segment-level consistency with multi-segment explicit domains. The representation enriched by multiple explicit domains bridges the gap between partially observed discrete samples and the underlying implicit temporal dynamics, enabling more accurate approximation of implicit temporal patterns for effective cross-domain adaptation. Our comprehensive evaluation across 6 time series benchmarks demonstrates EDEN's consistent superiority, achieving average accuracy improvements of 4.8% over state-of-the-art methods in cross-domain scenarios. Code is available at the anonymous link:https://github.com/mingyangliu1024/EDEN.

## 1 Introduction

Time series classification has been studied with immense interest in extensive applications and has made significant progress [15, 40, 37]. Nevertheless, practical deployment of the models often encounters severe performance degradation caused by distribution shifts between training and testing environments [27]. Unsupervised domain adaptation (UDA) [21, 12], leveraging knowledge transfer from a related source domain to the unlabeled target domain, can be a promising solution.

The core challenge of time series domain adaptation (TSDA) stems from the entanglement of domain shifts and intricate temporal patterns, which is further compounded by the partial observability of latent continuous-time dynamics. Unlike images or text, where raw observations capture rich semantic information, time series data constitutes an explicit sensor-limited single domain, whose discrete

39th Conference on Neural Information Processing Systems (NeurIPS 2025).

observations are constrained by acquisition parameters (e.g., sampling rates or record durations) that only partially reflect the underlying implicit domain of continuous-time processes [10], and hinder the modeling of intricate temporal patterns. Critically, modifying these sensor constraints yields different explicit domains, each emphasizing distinct aspects of the implicit temporal dynamics that must be effectively captured. Conventional approaches relying on a single explicit domain exhibit limited representational modeling capabilities, impeding domain invariant representation learning.

Current advanced methods primarily focus on extracting temporal representation directly from raw observational data [38, 25], demonstrating limited effectiveness in simultaneously capturing intricate temporal patterns and addressing their associated domain shifts. While recent advancements incorporating representations from frequency subspace show improved domain adaptation performance [14, 18], these approaches remain fundamentally constrained by fixed temporal-frequency integration paradigms. To unravel implicit temporal dynamics and enhance domain adaptation, it is essential to break through the limitations of the single view in the raw dataset.

In light of the above motivations, we propose **EDEN** (multiple **E**xplicit **D**omain **E**nhanced adaptation **N**etwork) for TSDA based on the integration of multiple explicit domains. By expanding the restrictions slightly, we expand the original dataset to (1) multi-scale explicit domains at fine-scale and coarse-scale, (2) multi-subspace explicit domains containing temporal subspace and frequency subspace, and (3) multi-segment explicit domains with nearby segments. Furthermore, EDEN investigates interactions among multiple explicit domains and achieves their effective integration with three coordinated modules: (1) For multi-scale explicit domains, we highlight that the coarse-scale features manifest smaller domain discrepancy, which is the metric proposed in domain adaptation theory [2]. Based on that, **Multi-Scale Curriculum Adaptation** is proposed to progressively align the source and target domain from coarse-scale to fine-scale. This curriculum learning strategy stabilizes global feature alignment before refining local discriminating details. (2) For multi-subspace explicit domains, we reveal that the discriminative capability of models for the same class may vary significantly between two subspaces. Based on that, we propose **Quality-Aware Feature Fusion**, an adaptive fusion mechanism that weighs subspace contributions based on their representation quality for specific instances. (3) For multi-segment explicit domains, we identify that nearby segments inherently possess similar class-related semantic information. Based on that, we propose **Temporal Coherence Learning**, encouraging the model to exhibit consistent and stable behavior on adjacent temporal windows. Main contributions are as follows:

1. Going beyond previous methods, we break through the sensor-limited single view and expand to multiple explicit domains, taking advantage of rich semantic information and comprehensive reflection of domain shift from multiple explicit domains, unraveling implicit temporal dynamics.

2. We propose EDEN, which integrates multiple explicit domains and enhances TSDA in three coordinate modules: Multi-Scale Curriculum Adaptation to align the source and target domain from coarse-scale to fine-scale; Quality-Aware Feature Fusion to adaptively integrate temporal-frequency features; Temporal Coherence Learning to encourage consistent and stable behavior.

3. EDEN achieves average accuracy improvements of 4.8% over state-of-the-art methods across a wide range of time series datasets in cross-domain scenarios.

## 2 Related Work

**General Unsupervised Domain Adaptation** Unsupervised domain adaptation leverages the labeled source domain to predict the labels of a different but related, unlabeled target domain. It has a wide range of applications [41, 42, 13]. To achieve this, UDA methods aim to minimize the domain discrepancy and thereby decrease the upper bound of the target error [2]. Existing UDA methods can be classified into three categories: (1) Methods based on adversarial training introduce a domain discriminator to distinguish source samples from target ones, while the feature extractor learns domain-invariant representations to fool the domain discriminator. Advanced methods include DANN [12], CDAN [22] and DIRT-T [31]. (2) Methods based on statistical divergence aim to extract transferable features by minimizing statistical domain discrepancy in a latent feature space. Widely used methods include DAN [21], DeepCoral [33] and HoMM [6]. (3) Methods based on self-training assign pseudo-labels on unlabeled target data and select confident samples to combine with source samples in the next iteration of training. Widely used methods include PFAN[7], CST [17] and AdaMatch [4]. Overall, these methods are generally designed. Although these methods can be

applied to time series through tailored feature extractors, they often yield suboptimal performance due to neglecting the unique characteristics of time series.

**Unsupervised Domain Adaptation for Time Series** To date, limited methods have been tailored to unsupervised domain adaptation for time series. Early works focus on modeling features in the temporal subspace. VRADA [26] and CoDATS [38] consider suitable feature extractors based on the temporal structure. SASA [5] adopts LSTM [30] to capture the domain-invariant association. AdvSKM [19] adapts MMD [34] to fit time series characteristics. CLUDA [25] learns contextual representation via contrastive learning. Recently, several works have highlighted the necessity of simultaneously modeling features in the temporal and frequency subspace for UDA. RAINCOAT [14] firstly introduces frequency features into domain adaptation, aligning temporal features and frequency features respectively via Sinkhorn divergence. ACON [18] proposes mutual learning and adversarial learning in temporal-frequency subspace. Despite remarkable progress, existing methods fail to exploit richer semantic in potential explicit domains, restricting domain shift mitigation.

# 3 Multiple Explicit Domains of Time Series

## 3.1 Problem Setup

In this paper, we study Unsupervised Domain Adaptation (UDA) problem for time series classification. Discrete time series are often a series of data points obtained by observing a continuous-time process at a discrete sequence of equally spaced points in time [10]. In time series classification problems, the dataset can be formalized as $D = \{(\mathbf{r}_i, \mathbf{y}_i)\}_{i=1}^n$, where $i$-th raw sample $\mathbf{r}_i \in \mathbb{R}^{C \times T}$ is sampled from $i$-th continuous-time process $\rho_i$, containing observation of $C$ variates over $T$ time steps.

In UDA setup, we are given $n_s$ raw labeled samples $\hat{P} = \{(\mathbf{r}_i^s, \mathbf{y}_i^s)\}_{i=1}^{n_s}$ drawn from the source distribution $P$ and $n_t$ raw unlabeled samples $\hat{Q} = \{(\mathbf{r}_i^t)\}_{i=1}^{n_t}$ drawn from the target distribution $Q$. Due to the domain shift between the source and target distribution, the model trained only on labeled source data encounters severe performance drops when deployed in the target domain. UDA for time series classification aims to learn a time series classification model with labeled source sample set $\hat{P}$ and unlabeled target sample set $\hat{Q}$, which can make accurate predictions on the target domain.

## 3.2 From Raw Time Series to Multiple Explicit Domains

Discrete time series datasets inherently include two explicit restrictions: sampling rate and record duration. The sampling rate determines how frequently observations are made, impacting the resolution and fidelity of captured data. The record duration defines the temporal extent, influencing the model's ability to capture trends and patterns within segments. Given $i$-th continuous-time process $\rho_i$, we obtain different samples by modifying the sampling rate and record duration. In mathematics, the continuous-time process set $\mathcal{P} = \{\rho_i\}_{i=1}^n$ forms an implicit domain, which is inaccessible due to sensor limitations. With explicit restrictions, the raw input set $\{\mathbf{r}_i\}_{i=1}^n$ forms an explicit domain.

Existing domain adaptation methods tailored for time series are mostly limited to a single explicit domain. By expanding the restrictions slightly, we treat frequency data as an explicit domain under the restriction of temporal-frequency transformation. To break through the limitations of the single view in the raw dataset and comprehensively reflect implicit temporal dynamics, we expand the original dataset into multiple explicit domains from three perspectives: record duration, sampling rate, and temporal-frequency transformation. As shown in Figure 1(a), given a raw time series dataset $\{\mathbf{r}_i\}_{i=1}^n$, for each instance $\mathbf{r}$, we expand it to three kinds of explicit domains:

**(1)** Given a raw sample $\mathbf{r}$, we segment it and obtain the set of segments $\{\mathbf{x}_j\}_{j=1}^K$, where $\mathbf{x}_j$ is a sub-series of $\mathbf{r}$ or $\mathbf{r}$ itself. The segments in $\{\mathbf{x}_j\}_{j=1}^K$ may have different length. After this step, each $\mathbf{r}$ in the original dataset is expanded to include $\{\mathbf{x}_j\}_{j=1}^K$ in the multi-segment explicit domains.

**(2)** Given a segment $\mathbf{x}_j$, we downsample $\mathbf{x}_j$ with a coarser scale $M$, and obtain the coarser-scale data $\mathbf{x}_j^c$. The segment $\mathbf{x}_j$ contains finer-scale information, denoted as $\mathbf{x}_j^f$. Each raw sample $\mathbf{r}$ is expanded to include $\{\mathbf{x}_j^f, \mathbf{x}_j^c\}_{j=1}^K$ in the multi-scale explicit domains.

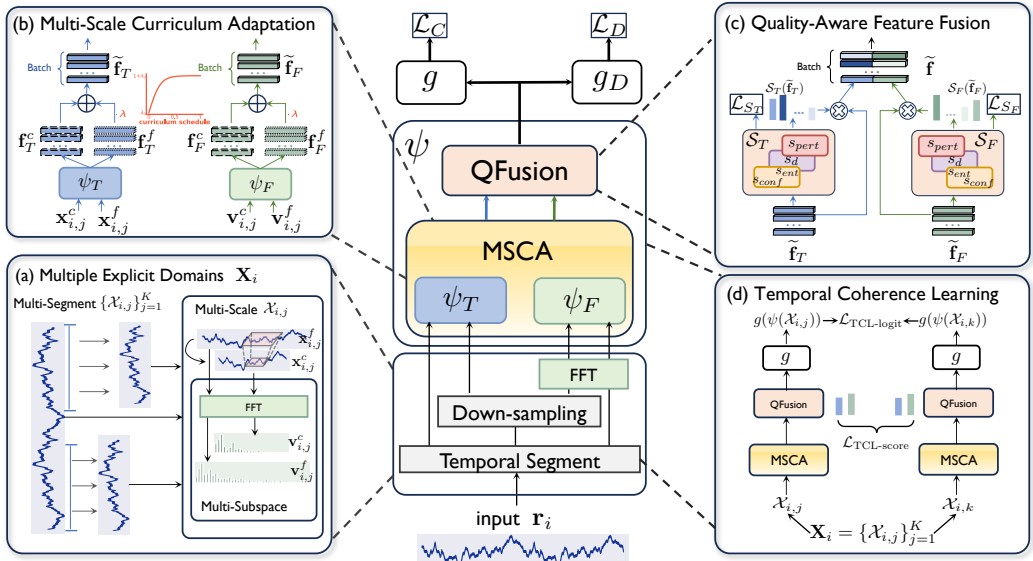

Figure 1: The schematic of EDEN, enhancing TSDA with expanded multiple explicit domains and three coordinated modules tailored to integrate multiple explicit domains. (a) Expanding the raw inputs into multiple explicit domains from three perspectives: record duration, sampling rate, and temporal-frequency transformation. (b) Multi-Scale Curriculum Adaptation implements progressive domain alignment through multi-scale explicit domains. (c) Quality-Aware Feature Fusion evaluates feature quality and adaptively integrates temporal-frequency features. (d) Temporal Coherence Learning enforces segment-level consistency constraints via multi-segment explicit domains.

(3) Given a fine-scale data $\mathbf{x}^f$ and corresponding coarse-scale data $\mathbf{x}^c$, we transform $\mathbf{x}^f$ and $\mathbf{x}^c$ into the frequency subspace by Fast Fourier Transform, resulting in complex variables $\mathbf{v}^f$ and $\mathbf{v}^c$. Each raw sample $\mathbf{r}$ is expanded to include $\{(\mathbf{x}_j^f, \mathbf{x}_j^c, \mathbf{v}_j^f, \mathbf{v}_j^c)\}_{j=1}^K$ in the multi-subspace explicit domains.

To simplify, **multiple explicit domains** $\{\mathbf{X}_i\}_{i=1}^n$ derived from raw dataset $\{\mathbf{r}_i\}_{i=1}^n$ are formalized as:

$$
\begin{aligned}
&\{\mathbf{X}_i\}_{i=1}^n, \mathbf{X}_i = \{\mathcal{X}_{i,j}\}_{j=1}^K, \\
&\mathcal{X}_{i,j} = (\mathbf{x}_{i,j}^f, \mathbf{x}_{i,j}^c, \mathbf{v}_{i,j}^f, \mathbf{v}_{i,j}^c).
\end{aligned}
\tag{1}
$$

The above three steps **do not alter the relevant category concepts**, and thus $\mathbf{X}_i$ share the same ground-truth label with $\mathbf{r}_i$. With the expanded multiple explicit domains, we have the labeled source domain $\hat{P} = \{(\mathbf{X}_i^s, \mathbf{y}_i^s)\}_{i=1}^{n_s}$ and the unlabeled target domain $\hat{Q} = \{(\mathbf{X}_i^t)\}_{i=1}^{n_t}$. Superscripts $s$ and $t$ are adopted to distinguish the source domain and the target domain.

## 4 Approach

Figure 1 illustrates the overall structure of EDEN, consisting of a temporal feature extractor $\psi_T$, a frequency feature extractor $\psi_F$, a domain discriminator $g_D$, a classifier $g$, and two auxiliary feature scorers $\mathcal{S}_T, \mathcal{S}_F$. Specifically, **(1)** To effectively utilize the multi-scale explicit domains, we propose Multi-Scale Curriculum Adaptation (MSCA) in Section 4.1. **(2)** To fully exploit the multi-subspace explicit domains, we propose Quality-Aware Feature Fusion (QFusion) in Section 4.2. **(3)** To utilize the multi-segment explicit domains, we propose Temporal Coherence Learning (TCL) in Section 4.3.

### 4.1 Multi-Scale Curriculum Adaptation

Intuitively, coarse-scale time series highlights macroscopic variations, i.e., low-frequency information, while deep neural networks demonstrate strong generalization capabilities for low-frequency information [39]. Fine-scale time series contain subtle changes in local regions, i.e., high-frequency information, which may enhance feature discriminability but pose substantial challenges for domain

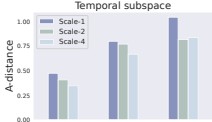 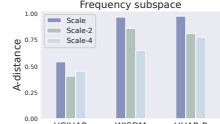 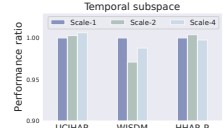 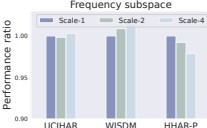

(a) $d_A$ in T-subspace    (b) $d_A$ in F-subspace    (c) DANN in T-subspace (d) DANN in F-subspace

Figure 2: Fine-scale vs. Coarse-scale: Denote the original sampling rate as $r_0$. Scale-$M$ refers to downsampling time series to a sampling rate of $\frac{r_0}{M}$. (a) A-distance in T-subspace. (b) A-distance in F-subspace. (c) DANN performance ratio of Scale-$M$ to original data (Scale-1) in T-subspace. (d) DANN performance ratio of Scale-$M$ to original data (Scale-1) in F-subspace. Reduced domain discrepancy may not guarantee accuracy gains due to the loss of discriminative information.

transfer. Transfer performance is jointly determined by transferability and discriminability. Therefore, multi-scale collaborative training may provide more performance gains compared to single-scale.

To validate the intuition, we investigate how representations with different scales influence the domain adaptation process. Based on the domain adaptation theory [2], the risk of the target domain can be bounded by the following proposition:

**Proposition 4.1** (Domain Adaptation Bound). *Let $\mathcal{H}$ be a hypothesis space, P, Q represent the source and target domain respectively. For every $h \in \mathcal{H}$, the target risk $\epsilon_Q(h)$ is bounded as:*

$$\epsilon_Q(h) \leq \epsilon_P(h) + \frac{1}{2}d_{\mathcal{H}\Delta\mathcal{H}(P,Q)} + \lambda^*, \tag{2}$$

*where $\epsilon_P(h)$ denotes the source risk, $\mathcal{H}\Delta\mathcal{H}$-distance $d_{\mathcal{H}\Delta\mathcal{H}(P,Q)} = 2\sup_{h,h'\in\mathcal{H}}|\epsilon_P(h,h') - \epsilon_Q(h,h')|$ measures domain shift as the discrepancy between the disagreement of two hypotheses $h, h'$, and $\lambda^* = \epsilon_P(h^*) + \epsilon_Q(h^*)$ is the error of the ideal joint hypothesis $h^*$.*

Under the supervision of source labels, $\epsilon_P(h)$ is usually smaller, while $\epsilon_Q(h)$ is mainly determined by the latter two terms. We adopt the proxy of $\mathcal{H}\Delta\mathcal{H}$-distance, A-distance [2], to quantify the domain discrepancy, defined as $d_A = 2(1-2\epsilon)$, where $\epsilon$ is the error rate of a domain classifier trained to discriminate source domain and target domain. As shown in Figure 2(a) and Figure 2(b), both in the temporal subspace and frequency subspace, it is consistently observed that the model trained on coarse-scale time series learns more domain-invariant features with smaller A-distance. However, as scale $M$ increases, the discriminative information gradually diminishes, potentially compromising prediction accuracy. In Figure 2(c) and Figure 2(d), we observe that reduced domain discrepancy may not guarantee accuracy gains with increasing $M$. This indicates that the coarse-scale features are easier for cross-domain transfer but potentially compromise discriminability. Consequently, multi-scale collaborative training emerges as a better choice.

Inspired by curriculum learning [9, 29, 3], we propose Multi-Scale Curriculum Adaptation to align the source and target distribution in an easy-to-hard way. In early training, coarse-scale features with stronger transferability guide the model to first stabilize global feature alignment. As the training progresses, fine-scale features gradually take the lead and refine local discriminating details.

Given multi-scale input $\mathcal{X} = (\mathbf{x}^f, \mathbf{x}^c, \mathbf{v}^f, \mathbf{v}^c)$, taking $(\mathbf{x}^f, \mathbf{x}^c)$ as the example, we extract fine-scale features and coarse-scale features respectively in the temporal subspace, i.e., $\mathbf{f}_T^f, \mathbf{f}_T^c = \psi_T(\mathbf{x}^f, \mathbf{x}^c)$ (similar to $(\mathbf{v}^f, \mathbf{v}^c)$), and mix the two features in a curriculum manner as follows:

$$\begin{aligned} \widetilde{\mathbf{f}}_T &= \lambda \mathbf{f}_T^f + (1 + 2\lambda_0 - \lambda)\mathbf{f}_T^c, \\ \lambda &= \frac{1 - \exp(-p)}{1 + \exp(-p)} + \lambda_0, \end{aligned} \tag{3}$$

where $\lambda$ is progressively increased, $p$ is the ratio of the current number to the maximum number of iterations, and the $\lambda_0$ is the initial value of $\lambda$. Denoting the mixing operation as $h_\lambda(\cdot)$, we extract the temporal feature $\widetilde{\mathbf{f}}_T = h_\lambda(\psi_T(\mathbf{x}^f, \mathbf{x}^c))$ and frequency feature $\widetilde{\mathbf{f}}_F = h_\lambda(\psi_F(\mathbf{v}^f, \mathbf{v}^c))$, and then concatenate them to obtain the final representation $\widetilde{\mathbf{f}}$. The progress is formulated as follows:

$$\psi(\mathcal{X}) = [h_\lambda(\psi_T(\mathbf{x}^f, \mathbf{x}^c)), h_\lambda(\psi_F(\mathbf{v}^f, \mathbf{v}^c))], \tag{4}$$

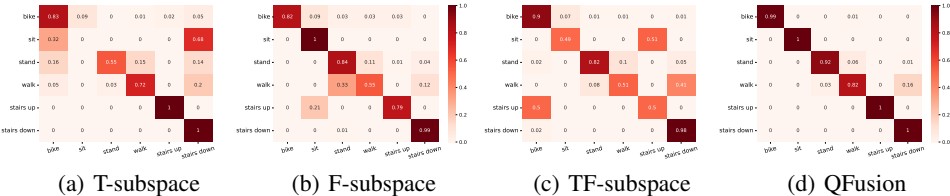

| (a) T-subspace | (b) F-subspace | (c) TF-subspace | (d) QFusion |
|---|---|---|---|

Figure 3: Error matrices: (a) Temporal subspace. (b) Frequency subspace. (c) Temporal-frequency subspace. (d)Temporal-frequency subspace with QFusion.

where $\psi$ is denoted as the holistic extractor containing $\psi_T$ and $\psi_F$. Based on the feature representation $\widetilde{\mathbf{f}} = \psi(\mathcal{X})$, we use the classifier $g$ to make the prediction $g(\psi(\mathcal{X}))$ and minimize $\mathcal{L}_C$ with the labeled source samples to guarantee lower source risk:

$$\mathcal{L}_C = \mathbb{E}_{(\mathbf{X}_i^s, \mathbf{y}_i^s) \sim \hat{P}, \mathcal{X}_{i,j}^s \sim \mathbf{X}_i^s} \left[ \mathrm{CE} \left( g \left( \psi \left( \mathcal{X}_{i,j}^s \right) \right), \mathbf{y}_i^s \right) \right], \tag{5}$$

where CE denotes cross-entropy loss. Meanwhile, we adopt the domain discriminator $g_D$ to align the source feature distribution and the target feature distribution via the minimax optimization problem:

$$\begin{aligned} \mathcal{L}_D = &-\mathbb{E}_{\mathbf{X}_i^s \sim \hat{P}, \mathcal{X}_{i,j}^s \sim \mathbf{X}_i^s} \log \left[ g_D \left( \psi \left( \mathcal{X}_{i,j}^s \right) \right) \right] \\ &- \mathbb{E}_{\mathbf{X}_i^t \sim \hat{Q}, \mathcal{X}_{i,j}^t \sim \mathbf{X}_i^t} \log \left[ 1 - g_D \left( \psi \left( \mathcal{X}_{i,j}^t \right) \right) \right], \end{aligned} \tag{6}$$

where $\mathcal{L}_D$ is minimized over $g_D$ but maximized over $\psi$.

### 4.2 Quality-Aware Feature Fusion

To explore the relationship between temporal subspace and frequency subspace, we delve into the error matrices of the target domain. We find that the classification performance of the same class may vary significantly between two subspaces. As shown in Figure 3(a) and Figure 3(b), the "sit" class is completely misclassified in the temporal subspace, while the class is classified correctly in the frequency subspace. Similarly, the "stairs up" class is classified correctly in the temporal subspace, but 21% of the class is incorrectly classified as "sits" in the frequency subspace. This suggests that the intrinsic characteristics of a certain class can be more effectively captured in the feature representation of one subspace compared to the other. In other words, each subspace has its advantageous classes.

Ideally, a model trained jointly in both subspaces, using the concatenation of temporal and frequency features as final features, would achieve optimal performance in each subspace for every class, which can classify both the "sit" class and the "stairs up" class correctly. However, as shown in Figure 3(c), the model does not actively select the best performance in the two subspaces; instead, it only achieves a compromise between the temporal and frequency subspace performance, or even worse.

This prompts us to seek a criterion for measuring the quality of features across different subspaces to achieve optimal subspace fusion. As shown in Figure 1(c), we introduce Quality-Aware Feature Fusion, which contains feature scorers $\mathcal{S}_T$, $\mathcal{S}_F$ into the temporal and frequency subspace respectively, responsible for measuring feature quality based on transferability, discriminability, and diversity, enabling adaptive temporal-frequency integration. With $\mathcal{S}_T$ and $\mathcal{S}_F$, Equation (4) is modified to:

$$\psi(\mathcal{X}) = [\mathcal{S}_T(\widetilde{\mathbf{f}}_T) \cdot \widetilde{\mathbf{f}}_T, \mathcal{S}_F(\widetilde{\mathbf{f}}_F) \cdot \widetilde{\mathbf{f}}_F]. \tag{7}$$

Each score $\mathcal{S}(\mathbf{f})$ is comprised of three components: transferability criterion, discriminability criterion and diversity perturbation. The first two are key criteria to characterize the performance of domain adaptation [8], while the latter ensures that different classes score diversely within the same subspace.

**Transferability Criterion** Transferability indicates the ability to learn domain-invariant features. Inspired by A-distance [2], a domain classifier is employed to measure transferability. The objective of the domain classifier is to predict source samples as 1 and target samples as 0. The closer the prediction $\hat{d}$ is to 0.5, the more difficult it is to distinguish the feature across domains, indicating

more domain-invariant. Thus, $\hat{d}$ can be used to quantify the transferability of each feature:

$$\hat{d} = \text{Sigmoid}\left(\text{MLP}(\mathbf{f})\right) \in [0, 1],$$
$$s_d = 2|\hat{d} - 0.5| \in [0, 1],$$

(8)

To train the domain classifier, $\ell_t = \text{CE}(\hat{d}, d)$ is calculated, where $d$ is the true domain label.

**Discriminability Criterion**    Discriminability is the easiness of separating different categories by a supervised classifier trained over the feature. Therefore, we use 1-layer MLP to classify the feature and the prediction $\hat{\mathbf{y}}$ contains the discriminative information about the feature. The entropy and confidence of $\hat{\mathbf{y}}$ are two uncertainty measurements on the discriminability. Smaller entropy, or higher confidence, means more discriminative features. Considering the complementarity of the two measurements [28], we use them simultaneously to quantify the discriminability of each feature:

$$\hat{\mathbf{y}} = \text{Softmax}\left(\text{MLP}(\mathbf{f})\right),$$
$$s_{ent} = 1 - H(\hat{\mathbf{y}}) \in [0, 1],$$
$$s_{conf} = \max\hat{\mathbf{y}} \in [0, 1],$$

(9)

where $H(\cdot)$ denotes the entropy, both $H(\hat{\mathbf{y}})$ and $\max\hat{\mathbf{y}}$ are min-max normalize. To train the MLP, $\ell_d = \text{CE}(\hat{\mathbf{y}}, \mathbf{y})$ is calculated on the source domain, where $\mathbf{y}$ is the ground-truth label of feature $\mathbf{f}$.

**Diversity Perturbation**    To enhance the diversity of scores within a subspace, we introduce diversity perturbation to facilitate an implicit competition between classes:

$$s_{pert} = \text{Sigmoid}\left(\text{MLP}(\mathbf{f})\right) \in [0, 1],$$
$$\ell_{cv} = \text{CV}(s_{pert}),$$

(10)

where $\text{CV}(\cdot)$ denotes the calculation of the coefficient of variation. The diversity perturbation encourages different classes to score diversely within the same subspace, especially promoting better scores for advantageous classes, thereby enhancing the discriminability of the features.

The overall score $\mathcal{S}(\mathbf{f})$ is formalized as follows:

$$\mathcal{S}(\mathbf{f}) = \frac{s_d + s_{ent} + s_{conf}}{3} + \eta s_{pert},$$

(11)

where $\eta$ controls the perturbation intensity and is set to 0.1 in all experiments. The training loss of the scorer $S$ is calculated as follows:

$$\ell_S = \ell_t + \ell_d + \ell_{cv}.$$

(12)

Given the heterogeneity of temporal and frequency subspace, we use $S_T$ and $S_F$ to measure their respective features independently, rather than sharing the same scorer. Consequently, we have two training losses, $\ell_{S_T}$ and $\ell_{S_F}$. The overall auxiliary loss is denoted as $\mathcal{L}_S = \ell_{S_T} + \ell_{S_F}$.

### 4.3   Temporal Coherence Learning

During the continuous-time process $\rho_i$, participants were required to adhere to a specific activity protocol to ensure the validity of the data collection [1]. Therefore, given the raw series $\mathbf{r}_i$, its segments share similar class-related semantics, e.g., consistently in a running or standing state.

Based on the intrinsic coherence of time series, we propose Temporal Coherence Learning. Given $\mathbf{X}_i = \{\mathcal{X}_{i,j}\}_{j=1}^K$, any $\mathcal{X}_{i,j}, \mathcal{X}_{i,k} \in \mathbf{X}_i$ that contain nearby segments exhibit temporal coherence, which can be reflected in logit-level and score-level. We achieve logit-level coherence by conditioning the logit distributions of $\mathcal{X}_{i,j}$ and $\mathcal{X}_{i,k}$ by minimizing the Kullback-Leibler divergence as follows:

$$\mathcal{L}_{\text{TCL-logit}} = \mathcal{D}_{KL}(g(\psi(\mathcal{X}_{i,j})), g(\psi(\mathcal{X}_{i,k}))).$$

(13)

To further complement the regularization, we impose score-level coherence learning and explicitly constrain the feature scores of QFusion assigned to nearby segments:

$$\mathcal{L}_{\text{TCL-score}} = |\mathcal{S}_T(\widetilde{\mathbf{f}}_{Tj}) - \mathcal{S}_T(\widetilde{\mathbf{f}}_{Tk})| + |\mathcal{S}_F(\widetilde{\mathbf{f}}_{Fj}) - \mathcal{S}_F(\widetilde{\mathbf{f}}_{Fk})|.$$

(14)

Overall, temporal coherence learning guides the feature extractors, scorers and classifier to exhibit consistent and stable behavior, especially in the unsupervised target domain.

$$\mathcal{L}_{\text{TCL}} = \mathcal{L}_{\text{TCL-score}} + \mathcal{L}_{\text{TCL-logit}}.$$

(15)

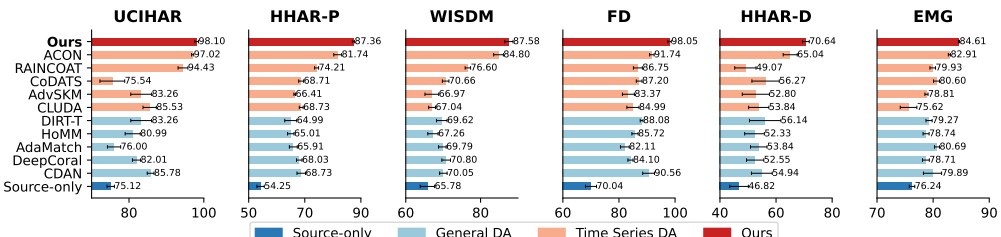

Figure 4: Average Accuracy (%) across multiple datasets.

Table 1: Accuracy (%) on EMG for unsupervised domain adaptation.

| Method | 0→1 | 0→2 | 0→3 | 1→2 | 1→3 | 2→0 | 2→1 | 2→3 | 3→1 | 3→2 | Average |
|---|---|---|---|---|---|---|---|---|---|---|---|
| Source-only | 84.94 | 74.38 | 73.38 | 74.38 | 73.88 | 73.88 | 82.16 | 73.69 | 79.38 | 72.31 | 76.24 |
| CDAN | 87.84 | 76.63 | 77.63 | 77.44 | 81.63 | 73.94 | 87.10 | 75.13 | 83.98 | 77.63 | 79.89 |
| DeepCoral | 87.50 | 76.44 | 76.19 | 77.63 | 77.63 | 74.69 | 84.72 | 75.50 | 81.93 | 74.88 | 78.71 |
| AdaMatch | 89.03 | 75.94 | 79.38 | 76.94 | 80.00 | 76.31 | 89.94 | 81.31 | 84.26 | 73.81 | 80.69 |
| HoMM | 87.61 | 76.50 | 75.75 | 77.00 | 77.94 | 73.94 | 84.89 | 75.88 | 82.61 | 75.31 | 78.74 |
| DIRT-T | 89.77 | 75.25 | 78.69 | 75.88 | 80.06 | 70.63 | 84.77 | 77.69 | 83.30 | 76.69 | 79.27 |
| CLUDA | 78.18 | 75.00 | 76.75 | 74.75 | 74.19 | 75.94 | 79.43 | 70.00 | 76.88 | 75.13 | 75.62 |
| AdvSKM | 86.42 | 75.94 | 76.25 | 77.25 | 78.00 | 74.88 | 85.06 | 77.25 | 81.76 | 75.31 | 78.81 |
| CoDATS | 88.24 | 77.44 | 78.31 | 78.44 | 81.81 | 73.75 | 86.65 | 78.88 | 84.43 | 78.06 | 80.60 |
| RAINCOAT | 89.60 | 77.00 | 78.56 | 78.25 | 83.13 | 73.06 | 85.68 | 76.88 | 83.13 | 74.00 | 79.93 |
| ACON | 92.50 | 79.06 | 81.75 | 80.13 | 83.13 | **77.94** | 90.91 | 79.75 | 85.11 | **78.88** | 82.91 |
| **Ours** | **93.92** | **82.31** | **83.38** | **81.19** | **86.50** | 77.12 | **92.78** | **82.88** | **87.61** | 78.38 | **84.61** |

## 4.4 Overall Training Objective

During training, our method trains the temporal feature extractor $\psi_T$, frequency feature extractor $\psi_F$ and classifier $g$ by minimizing the supervised classification loss $\mathcal{L}_C$ on the labeled source domain. To learn domain-invariant features, our method adopts the adversarial training facilitated with Gradient Reversal Layer between $\psi_T$, $\psi_F$ and the domain discriminator $g_D$ by minimizing the adversarial loss $\mathcal{L}_D$. Simultaneously, we train the feature scorers $\mathcal{S}_T$ and $\mathcal{S}_F$ to evaluate feature quality through the auxiliary loss $\mathcal{L}_S$. Furthermore, the coherence loss $\mathcal{L}_{\mathrm{TCL}}$ is minimized to preserve temporal coherence of $\psi_T$, $\psi_F$, $\mathcal{S}_T$, $\mathcal{S}_F$ and $g$. The overall training objectives is formulated as:

$$\mathcal{L} = \mathcal{L}_C + \mathcal{L}_D + \mathcal{L}_S + \beta\mathcal{L}_{\mathrm{TCL}}. \tag{16}$$

# 5 Experiments

## 5.1 Setup

**Datasets** We conduct extensive experiments using six benchmark datasets: (1) UCIHAR [1], HHAR-P [32, 27], WISDM[16] and HHAR-D [32, 11] for sensor-based human activity recognition. (2) FD [27] for machine fault diagnosis. (3) EMG [20, 23] for gesture recognition. For each dataset, following the existing DA methods on time series [4, 14, 18], we sample 10 source-target domain pairs for evaluation. Further details, processing and domain splits are included in Appendix A.

**Baselines** (1) We report the performance of the model without UDA (Source-only) in the temporal domain to show the overall contribution of UDA methods. (2) We implement the following state-of-the-art baselines for TSDA: CODATS [38], AdvSKM [19], CLUDA [25], RAINCOAT [14] and ACON [18]. (3) We additionally implement general UDA methods: CDAN [22], DeepCoral [33], AdaMatch [4], HoMM [6] and DIRT-T [31]. For each baseline, we ensure fair implementations by maintaining identical backbones, frameworks, and hyperparameter settings from prior works.

**Implementation** We adopt the implementation of AdaTime [27] as the benchmarking suites for domain adaptation on time series data. We use 1D-CNN as temporal feature extractor base and complex-valued linear as frequency feature extractor base. We report accuracy and Macro-F1 Score

 Each experiment is repeated 5 times with different random seeds. Detailed model architectures and optimal hyperparameters are included in Appendix B. Computational cost is included in Appendix C.1.

## 5.2 Results

Figure 4 shows the average accuracy and error bars of each method for 10 source-target domain pairs on the datasets. Overall, EDEN consistently improves the performance of the best baseline by a large margin. It's worth noting that EDEN also reduces the error bars, demonstrating the reliability. EDEN has won 6 out of 6 tests and makes an average improvement of 4.80% for the accuracy metric. Specifically, EDEN improves prediction accuracy by 6.87% on the HHAR-P dataset, 6.88% on the FD dataset, and 8.61% on the HHAR-D dataset over the advanced baseline on each dataset respectively. Due to the limited pages, we report the results for selected source-target domain pairs with metric accuracy on the EMG dataset. More accuracy results are given in Table 6-10. Average macro-f1 score is provided in Figure 8 and full macro-f1 score results are given in Table 12-17.

## 5.3 Analysis

**Module Ablation**  We conduct a comprehensive ablation study in Table 2. We investigate the effectiveness of three modules of EDEN and analyze the interaction between them. EDEN's three modules show performance gains when added individually (Rows 2-4) and demonstrate mutual promotion in pairwise combinations (Rows 5-7). Full integration (Row 8) achieves optimal performance.

Table 2: Ablation of EDEN, MSCA and QFusion.

| Module Ablation | | | | | | | |
|---|---|---|---|---|---|---|---|
| MSCA | QFusion | TCL | EDEN | UCIHAR | HHAR-P | WISDM | Average |
| - | - | - | - | 96.19 | 80.64 | 82.68 | 86.50 |
| ✓ | - | - | - | 97.16 | 83.47 | 83.58 | 88.07 |
| - | ✓ | - | - | 97.48 | 85.92 | 83.79 | 89.06 |
| - | - | ✓ | - | 97.21 | 82.14 | 84.04 | 87.80 |
| ✓ | - | ✓ | - | 97.65 | 85.75 | 85.53 | 89.64 |
| ✓ | ✓ | - | - | 97.77 | 86.12 | 85.43 | 89.77 |
| - | ✓ | ✓ | - | 97.88 | 85.97 | 85.74 | 89.86 |
| - | - | - | ✓ | **98.10** | **87.36** | **87.58** | **91.01** |
| MSCA Ablation | | | | | | | |
| $\psi_T$ | $\psi_F$ | $\mathcal{L}_D$ | MSCA | UCIHAR | HHAR-P | WISDM | Average |
| ✓ | - | - | - | 75.12 | 54.25 | 65.78 | 65.05 |
| ✓ | - | ✓ | - | 95.13 | 76.72 | 76.48 | 82.78 |
| ✓ | - | ✓ | ✓ | 96.17 | 77.55 | 81.10 | 84.94 |
| - | ✓ | - | - | 66.88 | 51.08 | 56.47 | 58.14 |
| - | ✓ | ✓ | - | 94.50 | 75.93 | 73.03 | 81.15 |
| - | ✓ | ✓ | ✓ | 96.37 | 80.97 | 76.05 | 84.46 |
| ✓ | ✓ | ✓ | ✓ | **97.16** | **83.47** | **83.58** | **88.07** |
| QFusion Ablation | | | | | | | |
| $s_d$ | $s_{ent}$ | $s_{conf}$ | $s_{pert}$ | UCIHAR | HHAR-P | WISDM | Average |
| - | - | - | - | 96.19 | 80.64 | 82.68 | 86.50 |
| ✓ | - | - | - | 97.00 | 83.05 | 82.77 | 87.61 |
| - | ✓ | ✓ | - | 97.42 | 84.32 | 82.94 | 88.23 |
| ✓ | ✓ | ✓ | ✓ | **97.48** | **85.92** | **83.79** | **89.06** |

**MSCA Ablation**  We verify the effectiveness of MSCA in the different subspaces. MSCA yields an average improvement of 2.61% in the temporal subspace (as shown in Rows 9-11 of Table 2), and an average improvement of 4.08% in the frequency subspace (Rows 12-14). By integrating the temporal subspace and the frequency subspace (Row 15), MSCA achieves the best performance, outperforming the single-subspace MSCA variants by 3.68%.

**QFusion Ablation**  We investigate the effectiveness of three components in QFusion score. Rows 17-18 of Table 2 apply transferability and discriminability criterion respectively, both showing performance gains. In Row 19, by combining transferability, discriminability, and diversity, QFusion achieves the best performance, demonstrating synergistic effect.

**Sensitivity Analysis**  The sensitivity analysis on the coarser scale $M$, the trade-off $\beta$ of $\mathcal{L}_{TCL}$ and the parameter $\lambda_0$ are included in Appendix C.2. As shown in Figure 5-7, EDEN exhibits stable performance across reasonable variations and consistently outperforms baselines on multiple datasets.

## 6 Conclusion

In this paper, we propose EDEN, a novel framework for TSDA based on multiple explicit domains, first breaking through the limitations of the sensor-limited single view in the original datasets, taking advantage of the comprehensive reflection of implicit temporal dynamics. Specifically, Multi-Scale Curriculum Adaptation is proposed to align the source and the target domain; Quality-Aware Feature Fusion is proposed to facilitate adaptive integration of temporal and frequency features; Temporal Coherence Learning is proposed to encourage the model to exhibit consistent and stable behavior. Notably, EDEN yields significant performance improvements on a wide range of time series datasets.

## Acknowledgements

This work was supported by the National Natural Science Foundation of China (62306085, 62206074 , 62476071, U23B2055, U24A20328), Shenzhen College Stability Support Plan (GXWD20231130151329002, GXWD20220811173233001), CCF-ALIMAMA TECH Kangaroo Fund (CCF-ALIMAMA OF 2025001), Guangdong Basic and Applied Basic Research Foundation (2025A1515012932, 2025A1515011732), Shenzhen Science and Technology Program (KQTD20240729102154066, ZDSYS20230626091203008).

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

# A  Datasets

## A.1  Detailed Statistics

We conduct extensive experiments using a wide range of time series datasets. The detailed statistics for each dataset is included in Table 3. For UCIHAR, HHAR-P, WISDM and FD datasets, we use the processed versions released by AdaTime [27]. For HHAR-D datasets, we use the processed versions released by WOODS [11]. For EMG dataset, we use the processed version released by DIVERSIFY [24].

Table 3: Summary of datasets.

| Dataset | Subjects | Channels | Length | Class | Total |
|---------|----------|----------|--------|-------|-------|
| UCIHAR  | 30       | 9        | 128    | 6     | 3290  |
| HHAR-P  | 9        | 3        | 128    | 6     | 17934 |
| WISDM   | 30       | 3        | 128    | 6     | 2070  |
| HHAR-D  | 5        | 6        | 500    | 6     | 13674 |
| EMG     | 4        | 8        | 200    | 6     | 6883  |
| FD      | 4        | 1        | 5120   | 3     | 10916 |

## A.2  Dataset Processing

Each domain of datasets is randomly divided into 80% training, and 20% testing. We follow Adatime [27], apply Z-score normalization to both the training and testing splits of the data, using the following equation:

$$x_i^{normalize} = \frac{x_i - x^{mean}}{x^{std}}, \quad i = 1, 2, \ldots, N \tag{17}$$

where $N = N_s$ for the source domain data and $N = N_t$ for the target domain data. Note that both the training and testing splits are normalized based on the training set statistics only.

# B  Experimental Details

## B.1  Model Architecture

Different from existing methods, our method simultaneously captures both coarse-scale and fine-scale feature representations by tailored feature extractors. Coarse-scale and fine-scale data typically exhibit varying lengths and contain distinct patterns, making it difficult to extract features simultaneously using a single CNN or MLP. Recent foundational models tailored for multi-scale in time series analysis generally employ separate feed-forward layers per scale followed by feature aggregation [36, 35]. Aligned with the TSDA benchmark's architectures [27, 18] that are based on 1D-CNN in temporal subspace and MLP in frequency subspace, we follow the structure and additionally incorporate a smaller CNN and MLP. In this way, we use a larger feature extractor to capture fine-scale features and a smaller feature extractor to obtain coarse-scale features. This configuration enables dedicated extraction of both fine-scale and coarse-scale characteristics.

Table 4: Key hyperparameters for EDEN.

| Hyperparameter | UCIHAR | HHAR-P | WISDM | HHAR-D | EMG   | FD   |
|----------------|--------|--------|-------|--------|-------|------|
| Epoch          | 50     | 50     | 50    | 50     | 50    | 50   |
| Batch Size     | 32     | 32     | 32    | 32     | 32    | 32   |
| Coarse Scale   | 2      | 4      | 4     | 2      | 4     | 4    |
| Learning Rate  | 0.01   | 0.001  | 0.003 | 0.001  | 0.001 | 0.01 |

## B.2  Multi-scale explicit domains

As mentioned in Section 4.1, limiting the input to a single scale makes determining the optimal scale for each dataset time-consuming. Therefore, we model both fine-scale and coarse-scale features simultaneously, leveraging the advantages of each. The fine-scale data refers to the raw data. Coarse-scale data is obtained by downsampling fine-scale data at scale $M$. The coarse scales $M$ chosen for different datasets are listed in Table 4. Sensitivity analysis of scale $M$ is presented in Figure 5.

The proposed MSCA consistently achieves performance improvements across viable scales $M$ in $\{2, 4, 6, 8\}$, demonstrating the robustness of scale selection.

### B.3 Multi-segment explicit domains

We introduce our segment strategy in two phases: training and testing. For the training phase, given raw data $\mathbf{r} \in \mathbb{R}^{C \times T}$, we randomly crop two short segments from the original dataset using a window size of $T/2$, while retaining the original segment. These segments form multi-segment explicit domains and are constrained for coherence through Temporal Coherence Learning. For the testing phase, we divide the original data into two halves using the same window size of $T/2$ to ensure consistency in each test. We predict for both segments separately, and the ensemble of these predictions is recorded as the final prediction result.

## C  Further Analysis

### C.1  Computational cost

Table 5: Training time and model performance on the FD dataset.

| FD | CDAN | Raincoat | ACON | EDEN |
|---|---|---|---|---|
| Accuracy | 90.56 | 86.75 | 91.74 | 98.05 |
| Training Time | 1.08h | 2.55h | 1.25h | 1.38h |

**Raincoat, ACON, and EDEN vs. CDAN**: Due to additional cues like time-frequency transformation, methods tailored for time series achieve performance improvements while increasing training time.

**EDEN vs. Raincoat**: EDEN not only significantly outperforms Raincoat but also reduces the training time. Raincoat has a longer training time due to its reconstruction-correction mechanism.

**EDEN vs. ACON**: Due to the introduction of multi-scale and multi-segment explicit domains, the training time of EDEN is slightly longer than ACON. However, considering the significant performance gains of EDEN (6.88%), the total training time is entirely acceptable.

### C.2  Sensitivity Analysis

**Sensitivity Analysis of Coarse Scale $M$**     As shown in Figure 5, our method exhibits stable performance on the multiple datasets across reasonable variations, with optimal performance at scales of 2 or 4. The optimal scale of different datasets is included in Table 4.

**Sensitivity Analysis of TCL trade-off $\beta$**     We investigate EDEN's sensitivity to the hyperparameter $\beta$ in Equation (16). As shown in Figure 6, EDEN remains stable within $0.2 \sim 1.4$, with the optimal average performance at the value of 1. Therefore, $\beta$ is fixed at 1 in all our experiments.

**Sensitivity Analysis of $\lambda_0$**     The hyperparameter $\lambda_0$ determines the dominant strength of coarse-scale at the initial training. Coarse-scale features dominate early alignment, with the weight decreasing from $1 + \lambda_0 \to \lambda_0$ while fine-scale features progressively intensify, with the weight increasing from $\lambda_0 \to 1 + \lambda_0$. In all our experiments, $\lambda_0$ is fixed at 0.5. As shown in Figure 7, EDEN exhibits smooth variations, demonstrating that our Multi-Scale Curriculum Adaptation achieves collaborative training of multi-scale explicit domains, consistently outperforming single-scale.

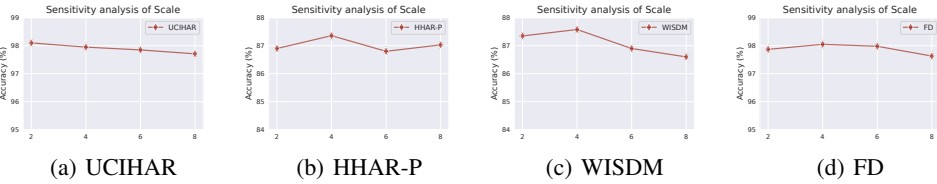

    (a) UCIHAR        (b) HHAR-P        (c) WISDM        (d) FD

Figure 5: Sensitivity Analysis of Coarse Scale $M$.

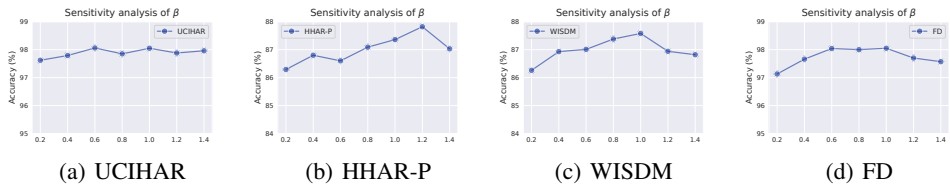

Figure 6: Sensitivity Analysis of TCL trade-off $\beta$.

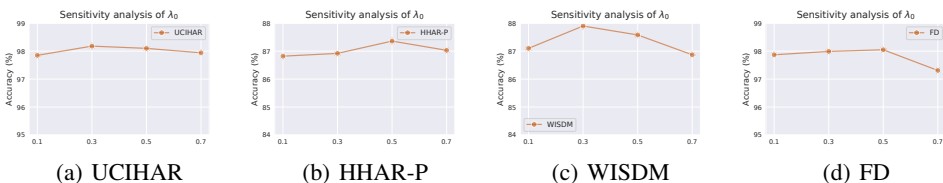

Figure 7: Sensitivity Analysis of $\lambda_0$.

# D Broader Impacts

We investigate the unsupervised domain adaptation method for time series classification by exploring multiple explicit domains. This paper aims to advance the field of time series analysis and the real-world deployment of time series applications without any negative social impact.

# E Full Results

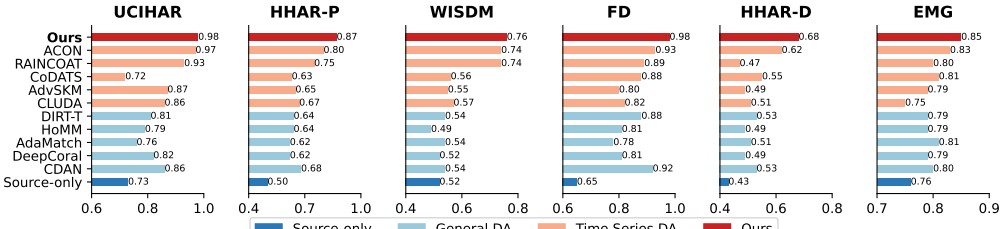

Figure 8: Average Macro-F1 Score across multiple datasets.

Table 6: Accuracy (%) on UCIHAR for unsupervised domain adaptation.

| Method | 2→11 | 6→23 | 7→13 | 9→18 | 12→16 | 13→19 | 18→21 | 20→6 | 23→13 | 24→12 | Average |
|---|---|---|---|---|---|---|---|---|---|---|---|
| Source-only | 76.56 | 67.36 | 83.68 | 24.65 | 61.11 | 88.89 | 100.0 | 94.10 | 71.18 | 83.68 | 75.12 |
| CDAN | 85.42 | 87.50 | 92.01 | 58.86 | 66.67 | 96.52 | 100.0 | 95.13 | 82.64 | 93.40 | 85.78 |
| DeepCoral | 90.63 | 84.38 | 87.50 | 46.88 | 65.28 | 95.49 | 100.0 | 95.49 | 69.79 | 87.50 | 82.01 |
| AdaMatch | 75.00 | 80.20 | 85.76 | 56.59 | 49.65 | 94.79 | 100.0 | 84.37 | 68.75 | 70.83 | 76.07 |
| HoMM | 74.06 | 82.71 | 81.88 | 73.96 | 70.21 | 96.67 | 98.75 | 73.33 | 77.71 | 80.63 | 80.99 |
| DIRT-T | 80.21 | 74.31 | 82.99 | 59.03 | 67.01 | 99.30 | 98.61 | 92.36 | 74.72 | 94.27 | 83.26 |
| CLUDA | 81.77 | 92.01 | 99.31 | 67.71 | 65.28 | 94.44 | 98.96 | 97.22 | 72.92 | 99.31 | 85.53 |
| AdvSKM | 98.96 | 88.54 | 92.71 | 74.65 | 69.44 | 93.05 | 100.0 | 85.41 | 79.51 | 96.87 | 83.26 |
| CoDATS | 68.23 | 74.31 | 77.43 | 63.89 | 66.32 | 94.09 | 99.65 | 70.49 | 56.25 | 82.81 | 75.54 |
| RAINCOAT | 100.0 | 95.83 | **100.0** | 75.69 | 86.52 | 100.0 | 100.0 | 93.41 | 86.52 | 93.75 | 94.43 |
| ACON | 100.0 | 96.25 | 99.16 | 91.66 | 85.63 | 100.0 | 100.0 | 97.50 | 100.0 | 100.0 | 97.02 |
| **Ours** | **100.0** | **99.17** | 98.96 | **93.12** | **90.62** | **100.0** | **100.0** | **99.17** | **100.0** | **100.0** | **98.10** |

Table 7: Accuracy (%) on HHAR-P for unsupervised domain adaptation.

| Method | 0→2 | 1→6 | 2→4 | 4→0 | 4→5 | 5→1 | 5→2 | 7→2 | 7→5 | 8→4 | Average |
|---|---|---|---|---|---|---|---|---|---|---|---|
| Source-only | 64.51 | 70.63 | 45.42 | 32.81 | 78.32 | 90.63 | 25.67 | 32.37 | 39.26 | 62.92 | 54.25 |
| CDAN | 76.19 | 92.57 | 52.57 | 29.09 | 97.27 | 96.16 | 35.04 | 37.05 | 75.26 | 96.11 | 68.73 |
| DeepCoral | 84.23 | 90.14 | 47.08 | 28.13 | 90.49 | 89.91 | 38.39 | 34.45 | 55.73 | 76.88 | 68.03 |
| AdaMatch | 84.78 | 92.31 | 54.50 | 36.45 | 78.45 | 94.20 | 41.96 | 37.65 | 63.80 | 64.69 | 65.91 |
| HoMM | 75.67 | 90.79 | 52.83 | 36.61 | 87.66 | 90.78 | 37.23 | 37.32 | 61.29 | 79.88 | 65.01 |
| DIRT-T | 77.83 | 88.54 | 50.69 | 32.22 | 93.16 | 91.86 | 38.62 | 38.10 | 72.46 | 65.83 | 64.99 |
| CLUDA | 79.84 | 93.40 | 45.90 | 38.84 | 94.08 | 95.57 | 33.93 | 37.80 | 77.57 | 96.52 | 69.35 |
| AdvSKM | 78.94 | 87.91 | 52.57 | 33.49 | 92.64 | 92.71 | 36.53 | 39.95 | 65.49 | 83.75 | 66.41 |
| CoDATS | 79.61 | 90.90 | 60.07 | 21.80 | 97.66 | 97.66 | 41.44 | 38.54 | 58.15 | 97.01 | 68.71 |
| RAINCOAT | 87.72 | 93.33 | 63.75 | 46.46 | 98.05 | 98.25 | 42.63 | 43.32 | 84.17 | 93.75 | 74.21 |
| ACON | 86.65 | 93.45 | 79.01 | 53.53 | 97.15 | 98.32 | **65.80** | 65.71 | 88.59 | 89.17 | 81.74 |
| **Ours** | **89.64** | **95.17** | **93.75** | **76.61** | **98.40** | **98.59** | 59.69 | **67.95** | **95.35** | **98.50** | **87.36** |

Table 8: Accuracy (%) on WISDM for unsupervised domain adaptation.

| Method | 2→32 | 4→15 | 7→30 | 12→7 | 12→19 | 18→20 | 20→30 | 21→31 | 25→29 | 26→2 | Average |
|---|---|---|---|---|---|---|---|---|---|---|---|
| Source-only | 81.16 | 79.86 | 89.32 | 71.53 | 54.29 | 83.74 | 67.96 | 21.29 | 26.11 | 82.52 | 65.78 |
| CDAN | 89.37 | 65.97 | 84.79 | 70.48 | 51.01 | 88.62 | 77.02 | 46.58 | 44.33 | 83.33 | 70.05 |
| DeepCoral | 87.92 | 62.50 | 91.26 | 79.86 | 51.77 | 64.23 | 81.88 | 54.62 | 53.89 | 77.44 | 70.80 |
| AdaMatch | 74.39 | 78.47 | 89.64 | 73.26 | 55.30 | 75.20 | 74.76 | 31.32 | 57.78 | 87.20 | 69.79 |
| HoMM | 77.10 | 74.58 | 78.64 | 68.13 | 50.61 | 71.22 | 72.82 | 56.39 | 57.00 | 66.10 | 67.26 |
| DIRT-T | 77.78 | 70.83 | 90.61 | 70.20 | 51.51 | 85.36 | 71.84 | 54.41 | 60.04 | 66.46 | 69.62 |
| CLUDA | 73.91 | 67.36 | 86.40 | 65.97 | 49.24 | 83.74 | 72.49 | 49.97 | 35.00 | 86.47 | 67.04 |
| AdvSKM | 70.83 | 95.85 | 93.85 | 77.08 | 47.47 | 81.30 | 21.28 | 44.45 | 74.79 | 74.95 | 66.97 |
| CoDATS | 77.29 | 70.83 | 83.20 | 70.17 | 47.47 | 76.01 | 82.85 | 52.61 | 53.89 | 83.29 | 70.66 |
| RAINCOAT | 79.71 | **97.91** | 91.28 | 89.80 | **85.00** | **92.23** | **91.66** | 59.09 | **82.97** | 83.50 | 76.60 |
| ACON | 89.86 | 86.25 | **98.06** | **98.13** | 77.73 | 83.66 | 91.26 | 63.61 | 60.00 | **99.51** | 84.80 |
| **Ours** | **92.50** | 90.62 | 97.29 | 95.42 | 82.97 | 88.44 | 90.42 | **76.56** | 65.00 | 96.56 | **87.58** |

Table 9: Accuracy (%) on FD for unsupervised domain adaptation.

| Method | 0→1 | 0→2 | 0→3 | 1→0 | 1→2 | 2→0 | 2→1 | 2→3 | 3→0 | 3→2 | Average |
|---|---|---|---|---|---|---|---|---|---|---|---|
| Source-only | 62.21 | 53.71 | 62.41 | 63.91 | 73.95 | 64.08 | 93.17 | 95.54 | 57.08 | 74.31 | 70.04 |
| CDAN | 91.29 | 71.83 | 90.13 | 96.50 | 90.09 | 83.10 | 99.38 | 99.98 | 95.40 | 87.95 | 90.56 |
| DeepCoral | 75.54 | 71.79 | 76.03 | 89.13 | 83.55 | 76.34 | 98.84 | 98.55 | 87.50 | 83.71 | 84.10 |
| AdaMatch | 67.81 | 55.38 | 62.88 | 92.21 | 98.57 | 79.08 | 89.96 | 90.40 | 87.23 | 97.57 | 82.11 |
| HoMM | 81.54 | 71.63 | 78.17 | 89.89 | 84.78 | 76.03 | 98.71 | 99.55 | 90.94 | 85.96 | 85.72 |
| DIRT-T | 75.94 | 70.85 | 76.36 | 98.10 | 90.27 | 81.92 | 100.0 | 99.98 | 97.06 | 90.29 | 88.08 |
| CLUDA | 90.47 | 82.63 | 88.68 | 89.06 | 92.23 | 61.92 | 93.91 | 90.80 | 82.01 | 78.17 | 84.99 |
| AdvSKM | 74.71 | 66.05 | 73.30 | 87.86 | 86.29 | 76.85 | 98.66 | 99.38 | 84.89 | 85.74 | 83.37 |
| CoDATS | 81.79 | 73.26 | 83.15 | 89.22 | 88.68 | 81.43 | 99.89 | 100.0 | 85.47 | 89.00 | 87.20 |
| RAINCOAT | 85.18 | 79.40 | 89.04 | 78.84 | 90.11 | 81.43 | 95.18 | 96.81 | 77.39 | 94.08 | 86.75 |
| ACON | 86.52 | 69.00 | 86.96 | 97.92 | 99.80 | 84.29 | 98.62 | 98.93 | 97.72 | 97.66 | 91.74 |
| **Ours** | **98.48** | **97.39** | **98.71** | **98.37** | **100.0** | **89.22** | **100.0** | **100.0** | **98.37** | **100.0** | **98.05** |

Table 10: Accuracy (%) on HHAR-D for unsupervised domain adaptation.

| Method | 0→1 | 0→2 | 0→3 | 0→4 | 1→0 | 1→3 | 1→4 | 2→1 | 3→4 | 4→1 | Average |
|---|---|---|---|---|---|---|---|---|---|---|---|
| Source-only | 65.48 | 33.59 | 31.71 | 39.79 | 34.69 | 44.83 | 49.54 | 38.17 | 86.17 | 44.23 | 46.82 |
| CDAN | 69.86 | 48.28 | 38.22 | 48.42 | 48.75 | 60.48 | 51.33 | 47.84 | 87.33 | 48.89 | 54.94 |
| DeepCoral | 68.94 | 42.88 | 40.67 | 47.96 | 35.63 | 55.31 | 56.21 | 44.71 | 87.25 | 45.96 | 52.55 |
| AdaMatch | 71.78 | 39.60 | 39.74 | 47.50 | 52.50 | 55.48 | 58.33 | 46.49 | 85.83 | 41.15 | 53.84 |
| HoMM | 69.66 | 40.51 | 39.16 | 50.42 | 35.94 | 55.02 | 57.13 | 42.36 | 86.79 | 46.35 | 52.33 |
| DIRT-T | 68.37 | 42.14 | 47.21 | 52.92 | 41.25 | 60.14 | 55.63 | 46.73 | 92.25 | 54.81 | 56.14 |
| CLUDA | 71.78 | 39.60 | 39.74 | 47.50 | 52.50 | 55.48 | 58.33 | 46.49 | 85.83 | 41.15 | 53.84 |
| AdvSKM | 67.93 | 40.71 | 40.19 | 47.33 | 37.19 | 55.65 | 59.54 | 42.69 | 87.46 | 49.33 | 52.80 |
| CoDATS | 72.50 | 43.35 | 50.79 | 45.50 | 58.44 | 62.24 | 54.54 | 40.14 | 89.63 | 45.53 | 56.27 |
| RAINCOAT | 74.47 | 36.52 | 48.82 | 35.29 | 51.25 | 41.49 | 41.50 | 34.28 | 88.58 | 38.46 | 49.07 |
| ACON | **77.50** | **61.36** | **54.69** | **65.46** | 69.38 | 71.30 | 62.13 | 50.10 | 93.63 | 44.86 | 65.04 |
| **Ours** | 75.67 | 54.44 | 51.11 | 61.04 | **79.38** | **89.59** | **76.38** | **58.75** | **94.88** | **65.19** | **70.64** |

Table 11: Accuracy (%) on EMG for unsupervised domain adaptation.

| Method | 0→1 | 0→2 | 0→3 | 1→2 | 1→3 | 2→0 | 2→1 | 2→3 | 3→1 | 3→2 | Average |
|---|---|---|---|---|---|---|---|---|---|---|---|
| Source-only | 84.94 | 74.38 | 73.38 | 74.38 | 73.88 | 73.88 | 82.16 | 73.69 | 79.38 | 72.31 | 76.24 |
| CDAN | 87.84 | 76.63 | 77.63 | 77.44 | 81.63 | 73.94 | 87.10 | 75.13 | 83.98 | 77.63 | 79.89 |
| DeepCoral | 87.50 | 76.44 | 76.19 | 77.63 | 77.63 | 74.69 | 84.72 | 75.50 | 81.93 | 74.88 | 78.71 |
| AdaMatch | 89.03 | 75.94 | 79.38 | 76.94 | 80.00 | 76.31 | 89.94 | 81.31 | 84.26 | 73.81 | 80.69 |
| HoMM | 87.61 | 76.50 | 75.75 | 77.00 | 77.94 | 73.94 | 84.89 | 75.88 | 82.61 | 75.31 | 78.74 |
| DIRT-T | 89.77 | 75.25 | 78.69 | 75.88 | 80.06 | 70.63 | 84.77 | 77.69 | 83.30 | 76.69 | 79.27 |
| CLUDA | 78.18 | 75.00 | 76.75 | 74.75 | 74.19 | 75.94 | 79.43 | 70.00 | 76.88 | 75.13 | 75.62 |
| AdvSKM | 86.42 | 75.94 | 76.25 | 77.25 | 78.00 | 74.88 | 85.06 | 77.25 | 81.76 | 75.31 | 78.81 |
| CoDATS | 88.24 | 77.44 | 78.31 | 78.44 | 81.81 | 73.75 | 86.65 | 78.88 | 84.43 | 78.06 | 80.60 |
| RAINCOAT | 89.60 | 77.00 | 78.56 | 78.25 | 83.13 | 73.06 | 85.68 | 76.88 | 83.13 | 74.00 | 79.93 |
| ACON | 92.50 | 79.06 | 81.75 | 80.13 | 83.13 | **77.94** | 90.91 | 79.75 | 85.11 | **78.88** | 82.91 |
| **Ours** | **93.92** | **82.31** | **83.38** | **81.19** | **86.50** | 77.12 | **92.78** | **82.88** | **87.61** | 78.38 | **84.61** |

Table 12: Macro-F1 Score on UCIHAR for unsupervised domain adaptation.

| Method | 2→11 | 6→23 | 7→13 | 9→18 | 12→16 | 13→19 | 18→21 | 20→6 | 23→13 | 24→12 | Average |
|---|---|---|---|---|---|---|---|---|---|---|---|
| Source-only | 0.69 | 0.63 | 0.84 | 0.17 | 0.58 | 0.91 | 1.00 | 0.94 | 0.71 | 0.84 | 0.73 |
| CDAN | 0.85 | 0.88 | 0.91 | 0.61 | 0.64 | 0.97 | 1.00 | 0.95 | 0.82 | 0.92 | 0.86 |
| DeepCoral | 0.91 | 0.81 | 0.87 | 0.44 | 0.65 | 0.95 | 1.00 | 0.95 | 0.70 | 0.88 | 0.82 |
| AdaMatch | 0.73 | 0.81 | 0.86 | 0.55 | 0.48 | 0.94 | 1.00 | 0.84 | 0.67 | 0.70 | 0.76 |
| HoMM | 0.73 | 0.78 | 0.81 | 0.69 | 0.69 | 0.96 | 0.99 | 0.71 | 0.75 | 0.78 | 0.79 |
| DIRT-T | 0.81 | 0.68 | 0.82 | 0.58 | 0.62 | 0.99 | 0.98 | 0.92 | 0.74 | 0.93 | 0.81 |
| CLUDA | 0.81 | 0.92 | 0.99 | 0.67 | 0.64 | 0.94 | 0.99 | 0.98 | 0.71 | 0.99 | 0.86 |
| AdvSKM | 0.99 | 0.87 | 0.92 | 0.73 | 0.68 | 0.93 | 1.00 | 0.84 | 0.77 | 0.96 | 0.87 |
| CoDATS | 0.66 | 0.71 | 0.78 | 0.60 | 0.64 | 0.93 | 0.99 | 0.65 | 0.54 | 0.81 | 0.72 |
| RAINCOAT | 1.00 | 0.96 | **1.00** | 0.76 | 0.86 | 1.00 | 1.00 | 0.94 | 0.86 | 0.94 | 0.93 |
| ACON | 1.00 | 0.97 | 0.99 | **0.91** | 0.86 | 1.00 | 1.00 | 0.98 | 1.00 | 1.00 | 0.97 |
| **Ours** | **1.00** | **0.99** | 0.99 | **0.94** | **0.92** | **1.00** | **1.00** | **0.99** | **1.00** | **1.00** | **0.98** |

Table 13: Macro-F1 Score on HHAR-P for unsupervised domain adaptation.

| Method | 0→2 | 1→6 | 2→4 | 4→0 | 4→5 | 5→1 | 5→2 | 7→2 | 7→5 | 8→4 | Average |
|---|---|---|---|---|---|---|---|---|---|---|---|
| Source-only | 0.60 | 0.64 | 0.32 | 0.29 | 0.78 | 0.90 | 0.19 | 0.31 | 0.36 | 0.58 | 0.50 |
| CDAN | 0.70 | 0.93 | 0.52 | 0.27 | 0.98 | 0.98 | 0.35 | 0.32 | 0.76 | 0.97 | 0.68 |
| DeepCoral | 0.86 | 0.91 | 0.45 | 0.26 | 0.90 | 0.90 | 0.36 | 0.32 | 0.50 | 0.73 | 0.62 |
| AdaMatch | 0.83 | 0.93 | 0.46 | 0.32 | 0.76 | 0.94 | 0.40 | 0.37 | 0.60 | 0.61 | 0.62 |
| HoMM | 0.70 | 0.91 | 0.45 | 0.37 | 0.88 | 0.91 | 0.34 | 0.40 | 0.61 | 0.79 | 0.64 |
| DIRT-T | 0.76 | 0.86 | 0.51 | 0.30 | 0.93 | 0.90 | 0.36 | 0.34 | 0.73 | 0.64 | 0.64 |
| CLUDA | 0.82 | 0.94 | 0.44 | 0.40 | 0.94 | 0.96 | 0.37 | 0.36 | 0.65 | 0.84 | 0.67 |
| AdvSKM | 0.72 | 0.88 | 0.44 | 0.33 | 0.93 | 0.92 | 0.35 | 0.41 | 0.64 | 0.83 | 0.65 |
| CoDATS | 0.73 | 0.90 | 0.46 | 0.20 | 0.96 | 0.94 | 0.41 | 0.36 | 0.59 | 0.95 | 0.63 |
| RAINCOAT | 0.87 | 0.93 | 0.59 | 0.45 | 0.98 | 0.98 | 0.41 | 0.44 | 0.86 | 0.94 | 0.75 |
| ACON | 0.86 | 0.93 | 0.74 | 0.52 | 0.97 | 0.98 | **0.62** | 0.65 | 0.89 | 0.89 | 0.80 |
| **Ours** | **0.88** | **0.95** | **0.94** | **0.76** | **0.98** | **0.99** | 0.59 | **0.67** | **0.95** | **0.98** | **0.87** |

Table 14: Macro-F1 Score on WISDM for unsupervised domain adaptation.

| Method | 2→32 | 4→15 | 7→30 | 12→7 | 12→19 | 18→20 | 20→30 | 21→31 | 25→29 | 26→2 | Average |
|---|---|---|---|---|---|---|---|---|---|---|---|
| Source-only | 0.68 | 0.52 | 0.77 | 0.53 | 0.36 | 0.81 | 0.56 | 0.10 | 0.15 | 0.69 | 0.52 |
| CDAN | 0.72 | 0.44 | 0.70 | 0.50 | 0.31 | 0.87 | 0.64 | 0.31 | 0.23 | 0.71 | 0.54 |
| DeepCoral | 0.71 | 0.42 | 0.85 | 0.67 | 0.35 | 0.63 | 0.67 | 0.27 | 0.25 | 0.64 | 0.52 |
| AdaMatch | 0.59 | 0.54 | 0.76 | 0.67 | 0.38 | 0.66 | 0.54 | 0.16 | 0.24 | 0.74 | 0.54 |
| HoMM | 0.63 | 0.42 | 0.62 | 0.55 | 0.39 | 0.63 | 0.60 | 0.30 | 0.26 | 0.54 | 0.49 |
| DIRT-T | 0.65 | 0.41 | 0.78 | 0.56 | 0.39 | 0.67 | 0.65 | 0.28 | 0.21 | 0.54 | 0.54 |
| CLUDA | 0.64 | 0.61 | 0.81 | 0.59 | 0.41 | 0.70 | 0.70 | 0.27 | 0.26 | 0.75 | 0.57 |
| AdvSKM | 0.61 | 0.55 | 0.84 | 0.53 | 0.35 | 0.71 | 0.61 | 0.28 | 0.28 | 0.55 | 0.55 |
| CoDATS | 0.66 | 0.41 | 0.75 | 0.62 | 0.37 | 0.76 | 0.72 | 0.30 | 0.30 | 0.70 | 0.56 |
| RAINCOAT | 0.68 | **0.98** | 0.86 | 0.72 | **0.78** | **0.92** | 0.87 | **0.43** | **0.44** | 0.75 | 0.74 |
| ACON | 0.81 | 0.65 | **0.99** | **1.00** | 0.63 | 0.76 | 0.87 | 0.36 | 0.28 | **1.00** | 0.74 |
| **Ours** | **0.88** | 0.76 | 0.96 | 0.91 | 0.68 | 0.85 | **0.87** | 0.42 | 0.31 | 0.94 | **0.76** |

Table 15: Macro-F1 Score on FD for unsupervised domain adaptation.

| Method | 0→1 | 0→2 | 0→3 | 1→0 | 1→2 | 2→0 | 2→1 | 2→3 | 3→0 | 3→2 | Average |
|---|---|---|---|---|---|---|---|---|---|---|---|
| Source-only | 0.41 | 0.33 | 0.41 | 0.65 | 0.77 | 0.64 | 0.95 | 0.97 | 0.59 | 0.78 | 0.65 |
| CDAN | 0.91 | 0.76 | 0.90 | 0.95 | 0.92 | 0.86 | 1.00 | 1.00 | 0.94 | 0.91 | 0.92 |
| DeepCoral | 0.61 | 0.62 | 0.62 | 0.90 | 0.87 | 0.77 | 0.99 | 0.99 | 0.89 | 0.88 | 0.81 |
| AdaMatch | 0.50 | 0.45 | 0.46 | 0.91 | 0.98 | 0.80 | 0.93 | 0.93 | 0.87 | 0.97 | 0.78 |
| HoMM | 0.61 | 0.52 | 0.62 | 0.91 | 0.88 | 0.78 | 0.99 | 1.00 | 0.91 | 0.89 | 0.81 |
| DIRT-T | 0.80 | 0.62 | 0.70 | 0.97 | 0.93 | 0.84 | 1.00 | 1.00 | 0.96 | 0.93 | 0.88 |
| CLUDA | 0.84 | 0.80 | 0.79 | 0.88 | 0.93 | 0.50 | 0.95 | 0.90 | 0.84 | 0.80 | 0.82 |
| AdvSKM | 0.55 | 0.54 | 0.57 | 0.89 | 0.89 | 0.76 | 0.99 | 1.00 | 0.87 | 0.89 | 0.80 |
| CoDATS | 0.80 | 0.69 | 0.87 | 0.90 | 0.92 | 0.86 | 1.00 | 1.00 | 0.87 | 0.92 | 0.88 |
| RAINCOAT | 0.89 | 0.84 | 0.92 | 0.81 | 0.92 | 0.85 | 0.96 | 0.98 | 0.81 | 0.94 | 0.89 |
| ACON | 0.86 | 0.75 | 0.89 | 0.96 | 1.00 | 0.88 | 0.99 | 0.99 | 0.96 | 0.98 | 0.93 |
| **Ours** | **0.98** | **0.98** | **0.99** | **0.97** | **1.00** | **0.92** | **1.00** | **1.00** | **0.97** | **1.00** | **0.98** |

Table 16: Macro-F1 Score on HHAR-D for unsupervised domain adaptation.

| Method | 0→1 | 0→2 | 0→3 | 0→4 | 1→0 | 1→3 | 1→4 | 2→1 | 3→4 | 4→1 | Average |
|---|---|---|---|---|---|---|---|---|---|---|---|
| Source-only | 0.61 | 0.27 | 0.25 | 0.33 | 0.44 | 0.43 | 0.46 | 0.32 | 0.85 | 0.38 | 0.43 |
| CDAN | 0.67 | 0.42 | 0.35 | 0.42 | 0.66 | 0.57 | 0.50 | 0.44 | 0.88 | 0.44 | 0.53 |
| DeepCoral | 0.65 | 0.34 | 0.33 | 0.40 | 0.48 | 0.53 | 0.53 | 0.39 | 0.86 | 0.41 | 0.49 |
| AdaMatch | 0.69 | 0.36 | 0.36 | 0.41 | 0.60 | 0.49 | 0.56 | 0.41 | 0.86 | 0.36 | 0.51 |
| HoMM | 0.66 | 0.33 | 0.31 | 0.41 | 0.47 | 0.52 | 0.53 | 0.37 | 0.86 | 0.42 | 0.49 |
| DIRT-T | 0.66 | 0.38 | 0.40 | 0.44 | 0.52 | 0.60 | 0.53 | 0.39 | 0.93 | 0.49 | 0.53 |
| CLUDA | 0.69 | 0.36 | 0.36 | 0.41 | 0.60 | 0.49 | 0.56 | 0.41 | 0.86 | 0.36 | 0.51 |
| AdvSKM | 0.63 | 0.32 | 0.31 | 0.38 | 0.46 | 0.54 | 0.56 | 0.36 | 0.86 | 0.44 | 0.49 |
| CoDATS | 0.71 | 0.38 | 0.44 | 0.39 | 0.70 | 0.61 | 0.53 | 0.38 | 0.90 | 0.44 | 0.55 |
| RAINCOAT | 0.72 | 0.32 | 0.42 | 0.32 | 0.56 | 0.39 | 0.38 | 0.31 | 0.89 | 0.35 | 0.47 |
| ACON | **0.76** | **0.53** | **0.49** | **0.56** | 0.81 | 0.67 | 0.59 | 0.44 | 0.93 | 0.41 | 0.62 |
| **Ours** | 0.73 | 0.49 | 0.45 | 0.54 | **0.82** | **0.89** | **0.76** | **0.55** | **0.95** | 0.61 | **0.68** |

Table 17: Macro-F1 Score on EMG for unsupervised domain adaptation.

| Method | 0→1 | 0→2 | 0→3 | 1→2 | 1→3 | 2→0 | 2→1 | 2→3 | 3→1 | 3→2 | Average |
|---|---|---|---|---|---|---|---|---|---|---|---|
| Source-only | 0.85 | 0.74 | 0.74 | 0.74 | 0.75 | 0.75 | 0.82 | 0.74 | 0.78 | 0.72 | 0.76 |
| CDAN | 0.88 | 0.77 | 0.78 | 0.78 | 0.82 | 0.74 | 0.87 | 0.76 | 0.84 | 0.78 | 0.80 |
| DeepCoral | 0.87 | 0.76 | 0.76 | 0.78 | 0.78 | 0.75 | 0.84 | 0.76 | 0.82 | 0.75 | 0.79 |
| AdaMatch | 0.89 | 0.76 | 0.79 | 0.77 | 0.80 | 0.76 | 0.90 | 0.81 | 0.84 | 0.74 | 0.81 |
| HoMM | 0.87 | 0.77 | 0.76 | 0.77 | 0.78 | 0.74 | 0.84 | 0.76 | 0.82 | 0.75 | 0.79 |
| DIRT-T | 0.90 | 0.75 | 0.79 | 0.76 | 0.80 | 0.71 | 0.84 | 0.78 | 0.83 | 0.77 | 0.79 |
| CLUDA | 0.78 | 0.75 | 0.77 | 0.75 | 0.74 | 0.76 | 0.79 | 0.70 | 0.75 | 0.75 | 0.75 |
| AdvSKM | 0.86 | 0.76 | 0.76 | 0.77 | 0.78 | 0.76 | 0.85 | 0.77 | 0.81 | 0.75 | 0.79 |
| CoDATS | 0.88 | 0.77 | 0.78 | 0.79 | 0.82 | 0.74 | 0.86 | 0.79 | 0.84 | 0.78 | 0.81 |
| RAINCOAT | 0.89 | 0.77 | 0.79 | 0.78 | 0.83 | 0.73 | 0.85 | 0.77 | 0.83 | 0.74 | 0.80 |
| ACON | 0.92 | 0.79 | 0.82 | 0.80 | 0.83 | **0.78** | 0.91 | 0.80 | 0.85 | 0.79 | 0.83 |
| **Ours** | **0.94** | **0.82** | **0.83** | **0.81** | **0.87** | 0.77 | **0.93** | **0.83** | **0.87** | 0.79 | **0.85** |

