# OpenReview forum: "Breakthrough Sensor-Limited Single View: Towards Implicit Temporal Dynamics for Time Series Domain Adaptation"
_NeurIPS.cc/2025/Conference — NeurIPS 2025 poster_

### Official Review · Reviewer_aFd7 · 2025-06-21

**Clarity:** 3
**Significance:** 3
**Originality:** 2
**Rating:** 4
**Confidence:** 3

**Summary:**

The paper proposes EDEN, a method for time series unsupervised domain adaptation (UDA) that introduces multiple explicit domains—multi-scale, multi-subspace, and multi-segment—and integrates them via three coordinated modules to better model implicit temporal dynamics under partial observability. The method is evaluated on six benchmarks and shows consistent improvements.

**Questions:**

1. Can EDEN be extended to other TSDA tasks such as forecasting or anomaly detection, beyond classification?
2. Is the combination of all three modules necessary, or could a simpler subset achieve comparable performance?
3. How does the notion of “explicit domains” differ fundamentally from existing multi-view or multi-scale learning frameworks?

**Ethical Concerns:**

["NO or VERY MINOR ethics concerns only"]

**Final Justification:**

The authors’ rebuttal and additional experiments have addressed most of the concerns. While the reviewer still feels that the novelty remains limited, and some evaluations could be stronger, the paper offers a clear and systematic approach. For these reasons, the reviewer has decided to raise the score to borderline accept.

**Limitations:**

1. The proposed modules are largely based on established ideas, with limited methodological novelty.
2. The concept of “explicit domains” is not rigorously defined and may overlap with prior terminology.
3. Performance gains, while consistent, are relatively modest, and stronger ablation studies are needed to validate design choices.
4. The method’s computational cost and scalability are not thoroughly analyzed, which may impact practical deployment.
5. Evaluation is limited to classification; generalization to other TSDA tasks remains untested.

**Paper Formatting Concerns:**

The paper format is ok.

**Quality:**

2

**Strengths And Weaknesses:**

Strengths:
1. The paper addresses an important and challenging problem in time series domain adaptation.
2. The overall writing is clear and well-structured.
3. The proposed framework is systematic and incorporates several components aimed at enhancing representation learning.
Weaknesses:
1. The novelty is somewhat limited: the proposed modules (curriculum learning, adaptive fusion, segment-level consistency) are composed of well-known ideas, adapted to the time series domain without clear methodological breakthroughs.
2. The use of "explicit domains" is largely terminological; it is unclear how fundamentally this differs from multi-scale or multi-view learning.
3. While the reported performance gains are consistent, they are relatively modest (4.8% on average), and the experimental section lacks strong ablations to clarify the contribution of each module.
4. The model architecture is relatively complex, but there is limited analysis of computational cost or inference efficiency, which may impact real-world applicability
5. The method is evaluated only on classification tasks; it's unclear whether the proposed approach generalizes to other important TSDA settings such as forecasting or anomaly detection.
6. The paper lacks a clear comparison with more recent time-series specific UDA baselines beyond general-purpose UDA methods.
7. The paper does not clearly justify why the combination of these three strategies is particularly effective or necessary, as opposed to using one or two of them.

---

> ### Author Rebuttal · Authors · 2025-07-31
>
> Thanks for the helpful feedback!
>
> ### **Q1: Modules are composed of well-known ideas with limited novelty**
>
> We appreciate reviewer's insightful feedback regarding novelty assessment. Many high-level ideas (curriculum learning and fusion mechanism) are broad research paradigms, as evidenced by surveys [1-3]. However, **novelties of methods mentioned in surveys [1-3] are recognized in top venues (NeurIPS, ICML, ICLR)**. Hence, **solely using high-level ideas cannot limit novelty; the novelty lies in how to design special curricula and fusion mechanisms for special scenarios**.
>
> **Similarly, novelty of EDEN lies in special designs that address unique challenges of TSDA.**
>
> #### **1. Theoretical Insights**
>
> In UDA theory (Eq. 2 in paper), existing TSDA methods fail to consider transferability ($d_{\mathcal{H}\Delta\mathcal{H}}$) and discriminability ($\lambda^*$) to design special curricula, fusion mechanisms and consistency regularization. EDEN bridge this gap.
>
> #### **2. Method-Specific Innovations**
>
> **MSCA: Theory-Inspired Curriculum Design**
>
> - Contrary to heuristic curricula, we propose the ***first coarse-to-fine scale curriculum*** to consider transferability and discriminability guided by TSDA theory.
>
> **Quality-Aware Feature Fusion: Theory-Inspired Fusion Mechanism**
>
> - Replaces heuristic fusion with a ***learnable feature scoring module*** based on transferability-discriminability trade-offs.
>
> **Temporal Coherence Learning: Consistency at Score-Level**
>
> - The first to leverage *TSDA theory-inspired scores* for consistency learning.
>
> [1] A survey on curriculum learning. TPAMI, 2021.
>
> [2] Curriculum learning: A survey. IJCV, 2022.
>
> [3] Deep multimodal data fusion. ACM computing surveys, 2024.
>
> ### **Q2: Difference between explicit domains and multi-scale or multi-view learning**
>
> "Explicit domains" emerged naturally from this context: temporal dynamics are inherently implicit, while time series data are constrained by human-controlled explicit conditions.
>
> **Difference from multi-scale or multi-view learning**: EDEN's core contribution lies not in obtaining explicit domains but rather in how to use generated multiple explicit domains to tackle the unique challenge in TSDA: the entanglement of domain shift and intricate temporal patterns. This is beyond existing multi-scale or multi-view learning methods.
>
> - From terminological perspective: different connotations
>
>   - multi-scale vs. explicit domains
>     Multi-scale is one of three types of explicit domains, emphasizing different sampling rates, but it cannot cover multi-subspace and multi-segment included in our explicit domains.
>   - multi-view vs. explicit domains
>     Multi-view learning refers to datasets that contain different modalities or types of data, such as an object described by text, video, and audio [1], where multiple views **already exist** within the dataset. In contrast, explicit domains emphasize being artificially **derived from original time series** through changes in sampling rate, record duration, and time-frequency transform.
>
> - From the perspective of representation learning: different purposes
>
>   - multi-scale vs. explicit domains
>     The existing multi-scale learning approaches in time series leverage inherent **multiperiodicity** of time series, averaging forecasting series across multiple scales to enhance overall performance [2]. In contrast, multi-scale in EDEN emphasizes the **transferability differences of different-scale time series**, revealing that coarse-scale features exhibit better cross-domain transferability. This is a unique challenge in UDA that general multi-scale learning cannot address.
>
>   - multi-view vs. explicit domains
>     Existing multi-view learning in time series is to achieve self-supervised learning by utilizing **the consistency between different augmented data** [3]. However, EDEN is designed based on **UDA theory insight**.
>
>     $\epsilon_{Q}(h)  \leq \epsilon_{P}(h)+ \frac{1}{2}d_{\mathcal{H}\Delta\mathcal{H}(P, Q)} + \lambda^*$
>
>     In UDA theory, $d_{\mathcal{H}\Delta\mathcal{H}(P, Q)}$ means transferability, while $\lambda^*$ means discriminability. Both jointly determine UDA performance. From perspectives of **transferability and discriminability**, EDEN explores the relationships between different scales, subspaces, and segments.
>
> [1] Deep multi-view learning methods: A review. *Neurocomputing*, 2021.
>
> [2] Timemixer: Decomposable multiscale mixing for time series forecasting. *ICLR*, 2024.
>
> [3] Multi-view Self-Supervised Contrastive Learning for Multivariate Time Series. *ACMMM*. 2024.
>
> ### **Q3: Performance gains are modest (4.8%).**
>
> For comparison, we calculate improvements of methods relative to their respective SOTA at that time. Compared to baselines, EDEN achieves comparable performance gains. However, as field develops, performance will gradually approach its limits, and the rate of absolute improvement will also slow down. To further illustrate, we demonstrate performance gains of five general UDA methods relative to SOTA at the time of their publication. It is evident that the proportion of gains becomes smaller. However, this does not detract from the fact that these works are representative.
>
> | CLUDA    | CODATS   | RAINCOAT | ACON       | EDEN     |
> | -------- | -------- | -------- | ---------- | -------- |
> | 1.0%     | 0.7%     | 5.0%     | 8.9%       | 4.8%     |
> | DANN [1] | CDAN [2] | MDD [3]  | GVB-GD [4] | DALN [5] |
> | 2.24%    | 1.39%    | 1.37%    | 0.45%      | 0.44%    |
>
> [1] Domain adaptation with conditional transferable components. *ICML*, 2016.
>
> [2] Conditional adversarial domain adaptation. *NeurIPS*, 2018.
>
> [3] Bridging Theory and Algorithm for Domain Adaptation. *ICML*, 2019.
>
> [4] Gradually Vanishing Bridge for Adversarial Domain Adaptation. *CVPR*, 2020.
>
> [5] Reusing the Task-specific Classifier as a Discriminator: Discriminator-free Adversarial Domain Adaptation. *CVPR*, 2022.
>
> ### **Q4: Lack strong ablations: clarify each module's contribution and why all three modules' combination is optimal**
>
> We respectfully note that this analysis is already provided in our submission: **In Table 2**, comprehensive ablation studies have been included,  containing **Module Ablation**, **MSCA Ablation** and **QFusion Ablation**.
>
> In Module Ablation, we **not only investigate the contribution of each module, but also analyze their interaction**.
>
> - Each module's contribution
>
>   As detailed in **Line 292~295** of our paper and **Row 1~4** of Table 2, EDEN’s three modules show performance gains when added individually: MSCA yields an average absolute accuracy gain of 1.57 across three datasets; QFusion improves accuracy by 2.56; and TCL enhances performance by 1.30.
>
> - Optimal combination
>
>   As detailed in **Line 292~295** of our paper and **Row 1~8** of Table 2, using one or two of these modules achieves suboptimal results, and full integration of the three modules (EDEN) achieves best performance.
>   Against MSCA+TCL, EDEN improves +1.37 accuracy gain; against MSCA+QFusion, EDEN improves 1.24; against QFusion+TCL, EDEN improves 1.15. Thus, integrating all three modules emerges as the best choice.
>
> ### **Q5: Limited computational cost analysis**
>
> In **Appendix D.1** of our paper, we present training time of EDEN on FD dataset.
>
> We additionally report GPU memory consumption, FLOPs (Floating Point Operations), and Parameters, where FLOPs quantifies overall computational complexity. Our method achieves comparable computational cost to SOTA baselines, while delivering significant performance enhancement.
>
> |                   | CDAN   | Raincoat | ACON   | EDEN   |
> | ----------------- | ------ | -------- | ------ | ------ |
> | Accuracy          | 90.56  | 86.75    | 91.74  | 98.05  |
> | Training Time (h) | 1.08   | 2.55     | 1.25   | 1.38   |
> | Memory (MiB)      | 715.6  | 859.6    | 725.6  | 789.3  |
> | FLOPs (MFLOPs)    | 467.7  | 123.2    | 233.8  | 235.3  |
> | params            | 228.6K | 768.6K   | 731.7K | 765.3K |
>
> ### **Q6: Evaluation is limited to classification**
>
> Forecasting constitutes a regression task, as is the mainstream approach (reconstruction) for anomaly detection. UDA problem for regression tasks remains unresolved.
>
> First, regarding the development of UDA, the target error bound of UDA theory holds exclusively for classification. Subsequently, many works have continued to improve classification UDA based on this bound [1, 2]. Occasionally, other fields, such as CV, have explored regression problems, but they have lost theoretical guarantees [3]. Currently, **there are few UDA methods that simultaneously address both DA classification and regression problems**. Researching on classification is the mainstream approach in DA field.
>
> Second, TSDA methods (e.g. baselines in our paper) universally assume classification tasks unless explicitly stated otherwise, and there are no methods that can simultaneously address both classification and regression issues.
>
> Our paper focuses on theoretically-grounded design. MSCA and QFusion are designed inspired by the target error bound of UDA theory, which applies to classification tasks and is not suitable for forecasting or anomaly detection.
>
> [1] A theory of learning from different domains. Machine learning, 2010.
>
> [2] Bridging theory and algorithm for domain adaptation. ICML, 2019.
>
> [3] Representation Subspace Distance for Domain Adaptation Regression. ICML, 2021.
>
> ### **Q7: Lack clear comparisons with more recent TSDA baselines**
>
> In experimental section, we select five representative TSDA baselines published in the last five years: ACON(24), RAINCOAT(23), CLUDA(22), AdvSKM(21), and CODATS(20). Among these, ACON is SOTA. In Related Work, we provide detailed introductions and comparisons of these works. In addition to these works, if the reviewer could provide specific baselines that would facilitate further comparison, we would greatly appreciate it.
>
> Please do let us know if you have any further questions.

---

> > ### Comment · Reviewer_aFd7 · 2025-08-07
> >
> > Thank you to the authors for the detailed rebuttal.
> >
> > The provided clarifications and additional experimental results address most of my key concerns. But I still have concerns about the novelty of the proposed method.
> >
> > Based on the authors' response, I choose to raise my score.

---

> > > ### Author Response · Authors · 2025-08-08
> > > **Sincere Thanks for Your Reassessment and Deeper Discussion about Novelty (Part 2)**
> > >
> > > 2. **The novelty of Quality-Aware Feature Fusion**
> > >
> > > Adaptive feature fusion, the adaptive combination of features from different layers or branches, is an omnipresent part of modern network architectures. Broadly speaking, any task involving multiple data modalities or requiring collaboration among feature extractors necessitates a feature fusion strategy to integrate diverse representations. Thus, adaptive feature fusion is **a necessary approach** dictated by specific data types or tasks, such as multimodal learning (demanding compressive consideration of multi-modal features), multi-task learning (needing the joint training of multiple feature extractors), or our problem (**requiring synergistic integration of temporal and frequency features**).
> > >
> > > In this case, novelty should not be determined by whether adaptive feature fusion is used, but rather **evaluated by how a specific adaptive feature fusion mechanism is designed for a specific problem**. Here, we list three representative fusion mechanisms:
> > >
> > > - SENet [8] employs **channel attention** to achieve adaptive fusion across feature maps, whose core idea is to enable networks to dynamically learn importance weights of feature channels.
> > > - Multi-gate Mixture-of-Experts (MMoE) [9] utilizes **gating mechanisms** to adaptively select and fuse features from multiple expert encoders.
> > > - Attentional Feature Fusion [10] leverages **multi-scale spatial attention** to achieve adaptive fusion of cross-source or multi-level features.
> > >
> > > Continuing to this day, novel and task-focused fusion mechanisms are being actively proposed in fields such as multi-modal learning [11, 12, 13], object detection [14], and semantic segmentation [15].  **While these works share similar high-level ideas (adaptive fusion), their special designs in how to fuse underpin their novelty and advance the fields.**
> > >
> > > Similarly, our Quality-Aware Feature Fusion introduces **a learnable feature scoring module based on transferability-discriminability trade-offs in UDA theory**, to adaptively fuse temporal and frequency features for TSDA tasks. This distinguishes itself from existing fusion mechanisms, constituting its core novelty.
> > >
> > > [8] Squeeze-and-excitation networks. In *CVPR*, 2018.
> > >
> > > [9] Modeling task relationships in multi-task learning with multi-gate mixture-of-experts. In *KDD*,  2018.
> > >
> > > [10] Attentional feature fusion. In *WACV*, 2021.
> > >
> > > [11] Multi-modal Gated Mixture of Local-to-Global Experts for Dynamic Image Fusion. In *ICCV*, 2023.
> > >
> > > [12] Coupled mamba: Enhanced multimodal fusion with coupled state space model. In *NeurIPS*, 2024.
> > >
> > > [13] Simvg: A simple framework for visual grounding with decoupled multi-modal fusion. In *NeurIPS*, 2024.
> > >
> > > [14] Query-based temporal fusion with explicit motion for 3d object detection. In *NeurIPS*, 2023.
> > >
> > > [15] 2 -S3Net: Attentive Feature Fusion with Adaptive Feature Selection for Sparse Semantic Segmentation Network. In *CVPR*, 2021.
> > >
> > > 3. **The novelty of Temporal Coherence Learning**
> > >
> > > From Fixmatch [16], which constrains consistency between weakly and strongly augmented samples, to contrastive learning [17], which emphasizes consistency among positive samples, constraining consistency between similar samples is a common regularization method.
> > >
> > > In our paper, Temporal Coherence Learning leverages the inherent semantic consistency of nearby segments in time-series data. It simultaneously imposes constraints at both the logits level and **the theory-inspired scores level of QFusion**, thus establishing its novelty and distinctiveness.
> > >
> > > [16] Fixmatch: Simplifying semi-supervised learning with consistency and confidence. In *NeurIPS*, 2020.
> > >
> > > [17] Supervised contrastive learning. In *NeurIPS*, 2020.
> > >
> > >
> > >
> > > **We deeply appreciate your valuable time and insightful review, particularly your constructive discussion regarding novelty and your openness to reassessing the score.** We sincerely treasure this opportunity for a deeper discussion. We acknowledge your perspective that curriculum learning, adaptive fusion, and consistency constraints are well-explored ideas. However, as we have discussed in our argument, **if a lack of novelty is judged solely based on the use of these high-level ideas, then the majority of subsequent research (including much groundbreaking work) will face the same criticism.**
> > >
> > > We contend that true novelty lies in the unique design to mitigate domain-specific challenges. Your feedback has prompted us to articulate this core argument more clearly. We would be happy to elaborate on the novelties of EDEN's design in the revised version of our paper.
> > >
> > > If you have any further concerns, please send them to us. We look forward to discussing with you to improve our work further.
> > >
> > > Best Regards,
> > >
> > > All Authors

---

> ### Author Response · Authors · 2025-08-04
> **Looking forward to your feedback**
>
> Dear Reviewer aFd7,
>
> We sincerely thank you for your invaluable insights and constructive feedback. We deeply appreciate this opportunity to address your concerns and refine our work.
>
> In our response, we have addressed the following queries:
>
> - Clarification on EDEN's novelty from the perspectives of **theoretical insights** and **method-specific innovations** (W1&L1 in the original review -> Q1 in our rebuttal).
> - Explanation of **the fundamental differences** between explicit domains and multi-scale or multi-view learning from the perspectives of the terminological perspective and representation learning (W2&Q3&L2 in the original review -> Q2 in our rebuttal).
> - Justification of compelling performance gain (W3&L3 in the original review -> Q3 in our rebuttal).
> - Clarification on **inclusion of strong ablations**, which not only directly demonstrates the contribution of each module but also proves that the full integration of the three modules is the optimal choice (W6&W7&Q2 in the original review -> Q4 in our rebuttal).
> - Supplementary analysis of the computational cost, which includes not only **the training time analysis** already presented in our paper but also additional analyses of **memory, FLOPs, and parameters** (W4&L4 in the original review -> Q5 in our rebuttal).
> - Justification of **classification-focused evaluation scope** from the perspectives of the development of UDA and the existing TSDA methods' evaluation protocol (W5&Q1&L5 in the original review -> Q6 in our rebuttal).
> - Clarification on comprehensive baseline comparison (W6 in the original review -> Q7 in our rebuttal).
>
> We hope our responses have adequately addressed your concerns. If you have any further concerns or questions, please do not hesitate to let us know, and we will be more than happy to address them promptly. Thanks for your recognition of our work and the thoughtful and constructive feedback again!
>
> Best Regards,
>
> All Authors

---

> ### Author Response · Authors · 2025-08-08
> **Sincere Thanks for Your Reassessment and Deeper Discussion about Novelty (Part 1)**
>
> Thank you for your dedicated time and effort in reviewing our paper. We sincerely appreciate your thoughtful follow-up and positive assessment.
>
> We understand that for senior machine learning researchers, the standard of novelty is high and is influenced by personal research taste. Therefore, it is understandable if there are remaining concerns regarding the novelty of our method. We are delighted that there is still an opportunity before the discussion to delve deeper into what constitutes the novelty of our paper.
>
> > Original Review: "The proposed modules are largely based on established ideas (**curriculum learning, adaptive fusion, segment-level consistency**), with limited methodological novelty."
>
> As you noted, our method is indeed inspired by well-known ideas (curriculum learning, adaptive fusion, and segment-level consistency). However, contrary to your view, we respectfully maintain that **utilizing such high-level ideas does not preclude novelty in our method**. We would like to present objective arguments demonstrating that our contributions about methodological novelty meet and exceed the standards of the top conferences.
>
> 1. **The novelty of Multi-Scale Curriculum Adaptation**
>
> As a general idea in Machine Learning, Curriculum Learning [1] was **ranked first in the Most Influential ICML Papers of 	2009** (https://resources.paperdigest.org/2024/05/most-influential-icml-papers-2024-05). We acknowledge that our paper may not match the groundbreaking impact of the first work to propose the general idea of curriculum learning, nor can it rival the most influential ICML 2009 papers in terms of historical significance.
>
> However, **based on the idea of curriculum learning, what specific problem to address and how to design a novel and effective curriculum can constitute sufficient novelty.** For this reason, over the past 16 years, hundreds of curriculum learning-inspired works have been published at top conferences like ICML, NeurIPS, and ICLR. These papers did not propose fundamental ideas comparable to curriculum learning. Instead, they **applied the ideas of curriculum learning to specific tasks, designing tailored curricula**, which were also recognized by the most prestigious venues. Here, we list three influential works:
>
> - Reference [2] published in CVPR 2020 with more than 700 citations: *CurricularFace: Adaptive Curriculum Learning Loss for Deep Face Recognition.*
>   - Propose **an adaptive curriculum learning strategy from easy samples to hard samples** for deep face recognition.
> - Reference [3] published in NeurIPS 2021 with more than 1000 citations: *FlexMatch: Boosting Semi-Supervised Learning with Curriculum Pseudo Labeling.*
>   - Propose **a curriculum learning approach from a lower threshold to a higher threshold** to leverage unlabeled data.
> - Reference [4] published in NeurIPS 2020 with more than 100 citations: *Safe Reinforcement Learning via Curriculum Induction.*
>   - Propose **a curriculum induction from soft reset interventions to hard reset interventions** to gradually relax safety constraints to approach the original environment.
>
> To this day, the concept of curriculum learning continues to thrive across diverse domains, inspiring emerging fields such as AI4Science [5], diffusion model training [6], and vision foundation model fine-tuning [7]. While these works are *largely based on the established idea of curriculum learning*, they nevertheless design distinct curricula to address specific challenges, constituting their primary novelty contribution.
>
> Similarly, the novelty of Multi-Scale Curriculum Adaptation stems from the proposed ***first coarse-to-fine scale curriculum*** to consider transferability and discriminability guided by TSDA theory, to the best of our knowledge, which remains unexplored in existing curriculum learning methods and TSDA methods.
>
>
>
> [1] Yoshua Bengio, Jérôme Louradour, Ronan Collobert, and Jason Weston. Curriculum learning. In *ICML*, 2009.
>
> [2] CurricularFace: Adaptive Curriculum Learning Loss for Deep Face Recognition. In *CVPR*, 2020.
>
> [3] FlexMatch: Boosting Semi-Supervised Learning with Curriculum Pseudo Labeling. In *NeurIPS*, 2021.
>
> [4] Safe Reinforcement Learning via Curriculum Induction. In *NeurIPS*, 2020.
>
> [5] Curriculum Learning for Biological Sequence Prediction: The Case of De Novo Peptide Sequencing. In *ICML*, 2025.
>
> [6] Denoising Task Difficulty-based Curriculum for Training Diffusion Models. In *ICLR*, 2025.
>
> [7] Curriculum Fine-tuning of Vision Foundation Model for Medical Image Classification Under Label Noise. In *NeurIPS*, 2024.

---

### Official Review · Reviewer_RHTJ · 2025-06-30

**Clarity:** 3
**Significance:** 3
**Originality:** 3
**Rating:** 4
**Confidence:** 4

**Summary:**

The paper proposes EDEN, a novel framework for domain adaptation (UDA) in time series classification. The key innovation lies in expanding the raw dataset into multiple explicit domains, i.e. multi-scale, multi-subspace, and multi-segment, to better capture implicit temporal dynamics. It integrates these domains through three coordinated modules: MSCA to align domains progressively from coarse to fine scales; QFusion to combine temporal and frequency features based on their discriminative and transferable qualities; TCL to enforce consistency across adjacent segments.

**Questions:**

1. **Generalization ability** - Could EDEN be evaluated on single-dataset time series classification (without domain shifts) to verify its generalized feature extraction capability?  Could EDEN be tested in few short learning experiments to show its robustness and generalization ability?
2. **Baseline comparison** - The representation learning is also an effective solution for domain adaptation in time series, e.g., TS-TCC and TimeMAE. The author might add these works to the related work discussion and compare them to validate EDEN’s competitiveness.
3. **Reference Typo** - Duplicate references in Lines 378 and 380.
4. Could you provide some visualizations of representations to show the gap between the source data and target data and how the EDEN to improve the representation learning in different modules, especially including representations in explicit domains (a), domain alignment (b), and feature fusion (c).

**Ethical Concerns:**

["NO or VERY MINOR ethics concerns only"]

**Final Justification:**

The authors’ thorough rebuttal is appreciated. As I was already positive about the paper at the time of submission, I will retain my positive rating.

**Limitations:**

yes.

**Paper Formatting Concerns:**

NA.

**Quality:**

3

**Strengths And Weaknesses:**

**Strengths**

1. EDEN explicitly models multi-scale, multi-subspace, and multi-segment domains in time series domain adaptation, addressing the limitations of single-view sensor data.
2. The integration of temporal-frequency features with adaptive fusion (QFusion) combines these features based on their discriminative and transferable qualities.
3. The paper provides a theoretical Proposition of the Domain Adaptation Bound and covers a wide range of time series datasets, outperforming baselines.

**Weaknesses**

1. **Scalability** - The author discusses the limitations of single-view sensor data, experiments lack validation in low-resource scenarios, e.g., only a few data points in the source dataset, like few-shot learning. Testing EDEN’s ability to leverage multiple domains to compensate for scarce source data would strengthen robustness claims.
2. Some time series representation learning works in the time series domain adaptation is not included in the related work and experiments, refer to question 2.
3. No visualizations illustrate how multi-domain alignment and time-frequency fusion improve representations. The author should provide some case studies to show the improvement in these designed modules.

---

> ### Author Rebuttal · Authors · 2025-07-31
>
> Thank you for your helpful feedback! We are grateful that the reviewer highlights our motivations, technical solution, and theoretical insights. To clarify, we summarize the original weaknesses or questions and adjust the order of the issues.
>
> ### **Q1: Validation in low-resource scenarios (few-shot)**
>
> Thank you for your suggestion regarding additional experiments under the setting of few-shot source data to further validate EDEN's robustness.
>
> - Modifications to the UDA setting
>
>   We have modified the original UDA setting to a few-shot UDA setting. Specifically, we retain only K samples per class from the original labeled source data, while keeping the unlabeled target data unchanged.
>   On the UCIHAR dataset, we conducted experiments with **5-shot, 10-shot, and 15-shot** settings.
>
> - Performance
>
>   As shown in the table, EDEN not only achieves excellent performance under the original UDA setting, but also exhibits strong robustness and generalization capabilities in the few-shot UDA scenarios.
>
> |             | 5-shot | 10-shot | 15-shot | ALL DATA |
> | ----------- | ------ | ------- | ------- | -------- |
> | Source-only | 20.67  | 38.06   | 40.19   | 75.12    |
> | RAINCOAT    | 24.65  | 39.69   | 47.36   | 94.43    |
> | ACON        | 18.36  | 54.55   | 63.77   | 97.02    |
> | EDEN        | 27.79  | 58.77   | 69.52   | 98.10    |
>
> ### **Q2: Comparisons with TS-TCC and TimeMAE**
>
> TS-TCC and TimeMAE are time series self-supervised representation learning methods. In the UDA setting, we first perform self-supervised learning on the feature extractor using labeled source domain data and unlabeled target domain data. Subsequently, we fine-tune the feature extractor and classifier using labeled source data to perform classification on the target domain. We conduct experiments on the UCIHAR dataset.
>
> | Method      | UCIHAR |
> | ----------- | ------ |
> | Source-only | 75.12  |
> | TS-TCC      | 79.74  |
> | TimeMAE     | 81.66  |
> | EDEN        | 98.10  |
>
> Compared to the Source-only method, these representation learning methods extract generalized features through self-supervised learning. However, since the fine-tuning of the classifier still relies on labeled source data, the model tends to overfit the source domain, unable to mitigate domain shift, resulting in relatively limited performance gain.
>
> In summary, these representation learning methods primarily focus on general representation learning capabilities, while EDEN, with strong general representation learning abilities, places more emphasis on transferable representation learning. We will include these discussions in the related work section and add these comparisons in the experiment section in the next version.
>
> ### **Q3: Single-dataset time series classification**
>
> Following the setups and data splits of TS-TCC and TimeMAE, we conduct single-dataset classification experiments on the UCIHAR dataset.
>
> - Modifications to EDEN
>
>   Since only a single domain exists without domain shifts, we make minor modifications to EDEN. Specifically, we remove the domain discriminator in MSCA while retaining the curriculum mixing of coarse and fine-scale features. Additionally, we eliminate the transferability criterion in QFusion, performing feature fusion solely based on the discriminability criterion.
>
> - Performance
>
>   EEDEN achieves comparable experimental results in single-dataset time series classification compared to the advanced representation learning method TimeMAE and the general time series analysis method TimesNet [1], verifying its generalized feature extraction capability.
>
>   | Method   | Accuracy   |
>   | -------- | ---------- |
>   | TS-TCC   | 89.22±0.19 |
>   | TimeMAE  | 95.11±0.18 |
>   | EDEN     | 94.25±0.28 |
>   | TimesNet | 96.93±0.24 |
>
>   [1] Timesnet: Temporal 2d-variation modeling for general time series analysis. In *ICLR*, 2023.
>
> ### **Q4: Visualizations of domain shift**
>
> We adopt **t-SNE** to visualize the feature distributions on the UCIHAR dataset. However, **due to the limitations of the rebuttal format**, we're unable to upload images or external links. We can only provide **the Euclidean distances** $d$​ between the source and target features reduced to 2D space via t-SNE to show the domain gap.
>
> The second column of the table represents the gap between the centers of the two domains, **reflecting the discrepancy between the feature marginal distributions**. The third column indicates the average gap between the centers of each class in the two domains, **reflecting the discrepancy between the feature conditional distributions**.
>
> |                   | $d$ between two domain centers | average $d$ between class centers |
> | ----------------- | ------------------------------ | --------------------------------- |
> | Source-only       | 3.89                           | 3.53                              |
> | with MSCA         | 1.74                           | 2.87                              |
> | with MSCA+QFusion | 1.31                           | 1.55                              |
> | with EDEN         | 1.02                           | 1.26                              |
>
> After applying MSCA to align the distributions, we observe a significant reduction in the discrepancy of marginal distributions, demonstrating that **MSCA effectively aligns the marginal distributions of the two domains**. However, a considerable gap in conditional distributions still remains.
>
> When we add QFusion, we observe a substantial reduction in the discrepancy in conditional distributions. This validates the motivation behind QFusion and confirms the fusion mechanism's effectiveness. Specifically, the feature quality of the same class varies significantly across different subspaces. By using transferability and discriminability criteria, we can adaptively select higher-quality features, thus **achieving better alignment at the class level**.
>
>
>
> ### **Q5: Duplicate references in Lines 378 and 380**
>
> Thank you for pointing out this error. We will correct it in the next version.
>
>
>
> We hope our responses were able to address any remaining concerns. Please do let us know if you have any further questions as well as what would be expected for score improvement. Even a slight increase in score would greatly help in recognizing the potential of our work. Thank you again for providing insightful comments.

---

> > ### Author Response · Authors · 2025-08-04
> > **Looking forward to your feedback**
> >
> > Dear Reviewer RHTJ,
> >
> > Your comments have significantly contributed to strengthening our work, and we deeply appreciate the time and effort you dedicated to reviewing our paper.  We treasure the opportunity to address your concerns and improve our work.
> >
> > In our response, we have addressed the following queries:
> >
> > - We expand the experiments under the **few-shot UDA** (W1&Q1 in the original review -> Q1 in our rebuttal) and **single dataset classification** (Q1 in the original review -> Q3 in our rebuttal) settings to validate EDEN's generalized feature extraction capability and robustness.
> > - We add **two time series representation learning works** (W2&Q2 in the original review -> Q2 in our rebuttal) as baselines and provide a comparison with EDEN.
> > - We provide **visualization** (W3&Q4 in the original review -> Q4 in our rebuttal) comparisons before and after alignment, demonstrating the visualization results through **specific quantitative metrics**.
> >
> > We hope our response can address your concerns. If you have any further concerns or questions, please do not hesitate to let us know, and we will be more than happy to address them promptly. Thanks for your recognition of our work and the thoughtful and constructive feedback again!
> >
> > Best Regards,
> >
> > All Authors

---

> > ### Comment · Reviewer_RHTJ · 2025-08-05
> >
> > Thank you to the authors for your thorough response. Since I was already positive about the paper at the time of submission, I will retain my positive rating.

---

> > > ### Author Response · Authors · 2025-08-05
> > >
> > > Thank you for your invaluable feedback and dedicated time to review our paper.
> > >
> > > We will revise our paper according to your feedback, adding additional experiments under the few-shot UDA and single dataset classification settings, including the time series representation learning works as baselines, and providing visualization to demonstrate the improvement in these designed modules.
> > >
> > > If you have any further questions, please send them to us. We look forward to discussing with you to further improve our work.

---

### Official Review · Reviewer_KFEc · 2025-07-04

**Clarity:** 2
**Significance:** 2
**Originality:** 3
**Rating:** 4
**Confidence:** 1

**Summary:**

This paper find a unique perspective of TSDA and therefore proposes three strategies to model this orginal dataset, i.e., multi-scale, multi-space and multi-segment modeling. The exp results seems ok for me.

**Questions:**

1）Curriculum Learning Justification
The paper assumes coarse-scale features are easier and more domain-invariant, motivating the coarse-to-fine curriculum. However, this assumption needs clearer theoretical or empirical support.

2）Handling Different Time Scales
The method assumes source and target share at least comparable temporal resolutions. It is unclear how the model performs when the target has a very different time scale. Clarification or discussion of robustness in such scenarios is needed.

3）Limited Explicit Domain Perspectives
Only three types of explicit domains are explored. Could other forms—e.g., modality, condition, or context—be useful? A brief discussion on unexplored directions would improve completeness.

4）Module Naming Consistency
The module names are somewhat inconsistent. Consider the following renames to better match Multi-Scale Curriculum Adaptation and improve conceptual clarity.

Quality-Aware Feature Fusion → Quality-Aware Multi-Subspace Attention

Temporal Coherence Learning → Coherence-Aware Multi-Segment Alignment

**Ethical Concerns:**

["NO or VERY MINOR ethics concerns only"]

**Limitations:**

yes

**Paper Formatting Concerns:**

no formatting concern

**Quality:**

2

**Strengths And Weaknesses:**

Strengths:

1）The overall motivation is sound. Since different time series data can exhibit diverse explicit patterns, modeling them from multiple perspectives is a reasonable and compelling direction.

2）The results are promising, achieving state-of-the-art performance compared to a recent NeurIPS 2024 method.

Weaknesses:

1）The abstract and introduction lack clarity in conveying the core ideas behind "multi-scale" and "multi-subspace" modeling. For instance, the role and mechanism of Quality-Aware Feature Fusion and Multi-Scale Curriculum Adaptation could be summarized more clearly. Currently, I have to dive into the full method section to grasp how it works. A concise explanation or intuitive example early on would significantly improve readability.

---

> ### Author Rebuttal · Authors · 2025-07-31
>
> Thank you for your helpful feedback! We are grateful that the reviewer highlights our motivations and experimental results.
>
> ### **W1: The abstract and introduction lack clarity in conveying the core ideas behind "multi-scale" and "multi-subspace" modeling.**
>
> Thank you for your valuable feedback.
>
> We will revise the abstract and introduction to improve clarity and add explanations:
>
> - "Multi-Scale Curriculum Adaptation progressively integrates coarse and fine-scale features to obtain the final extracted features, facilitating curriculum adaptation from coarse-to-fine by aligning the mixed feature between source and target domains. "
> - "Quality-Aware Feature Fusion measures feature quality through transferability and discriminative scores for adaptive fusion. The transferability criterion is determined by A-distance, which reflects the prediction confidence of the domain classifier, while the discriminative criterion is based on the entropy and confidence of the classifier. To enhance score diversity at the class level, we also introduce diversity perturbation by encouraging the coefficient of variation."
>
> ### **Q1: The assumption that coarse-scale features are easier and more domain-invariant needs clearer theoretical or empirical support.**
>
> Empirically, in **Figure 2 (a)(b)** of our paper, we **use A-distance [1] to quantify domain discrepancy** of features at different scales. We further report the A-distance on three additional datasets in the following table. Combined with Figure 2(a)(b), the results validate our assumption on all six datasets, demonstrating that aligned features of coarse-scale time series exhibit smaller domain discrepancies. This quantitative analysis adequately supports the proposed coarse-to-fine curriculum.
>
> Theoretically, as noted in Proposition 4.1 of our paper, A-distance $d_A = 2(1-2\epsilon)$is determined by the generalization error $\epsilon
> $ of discriminating between source and target examples, and serves as a proxy of $\mathcal{H}$-divergence [2] between the source domain $P$ and target domain $Q$ , $d_\mathcal{H}(P, Q)=2\sup_{h\in\mathcal{H}}|Pr_P[h]-Pr_Q[h]|$​. This metric guides the family of domain adversarial learning methods (e.g., DANN, CDAN), providing valid estimates of domain discrepancy.
>
> | A-distance | FD   | HHAR-D | EMG  |
> | :--------- | :--- | :----- | :--- |
> | Scale 1    | 0.18 | 0.82   | 0.36 |
> | Scale 2    | 0.13 | 0.80   | 0.35 |
> | Scale 4    | 0.11 | 0.68   | 0.31 |
>
> [1] Domain-Adversarial Training of Neural Networks. *JMLR*, 2016.
>
> [2] A theory of learning from different domains. *Machine learning*, 2010.
>
> ### **Q2: Handle different time scales of source and target data**
>
> In existing time series analysis work, it is generally assumed that **all training and testing data for the same feature extractor should have the same resolution**. When there is a significant difference in resolution among data, it is typically addressed by ensuring consistency through downsampling. This is because time series data has a high information density; when the sampling rates differ, a segment of length 5 may capture variations within 5 hours in one case and variations over 5 days in another, leading to completely different distributions. Therefore, it is not feasible to use the same feature extractor for both.
>
> ### **Q3: Only three types of explicit domains are explored. other forms—e.g., modality, condition, or context—be useful?**
>
> Thank you for your valuable insights. Leveraging other forms for domain adaptation is a promising research direction.
> However, it is important to note that the other forms, e.g., modality, condition, or context, typically **depend on external knowledge**, which is not universally available across datasets.
>
> In contrast, our proposed three explicit domains are derived from the original time series **without relying on any external information**, and are universal across multiple datasets. These explicit domains are the only three explicit domains, which is fundamentally determined by the intrinsic limitations of time series sampling (sampling rate, record duration) and the inherent property that discrete signals can be transformed into the frequency domain.
>
> We believe that external modalities, conditions, or contexts can enhance UDA performance from two aspects.
>
> - Meta-information guided transfer
>   For example, in action recognition tasks, we know the context of the subjects' dominant hands—one being left-handed and the other right-handed. Using this information, we can adjust the distribution of channels in the original series to achieve spatial alignment at the channel level.
> - External conditions assisted classification
>   For example, in action recognition tasks, the data not only includes time series but also contains determining conditions. For instance, if the main frequency is below a specified value, it indicates a stationary state. We can use this information to assist in classification.
>
> ### **Q4: Module Naming Consistency**
>
> Thank you for your suggestion. We will revise the module names in the next version to ensure naming consistency.
>
>
>
>
>
>
> We hope our responses were able to address any remaining concerns. Please do let us know if you have any further questions as well as what would be expected for score improvement. Even a slight increase in score would greatly help in recognizing the potential of our work. Thank you again for providing insightful comments.

---

> > ### Author Response · Authors · 2025-08-04
> > **Looking forward to your feedback**
> >
> > Dear Reviewer KFEc,
> >
> > Thank you very much for your insightful and valuable comments! We treasure the opportunity to address your concerns and improve our work.
> >
> > In our response, we have addressed the following queries:
> >
> > - Improvement of readability in the abstract and introduction (W1).
> > - The theoretical and empirical support for curriculum learning justification (Q1).
> > - Clarification about target data with different scales (Q2).
> > - Discussion about other forms of explicit domains (Q3).
> > - Enhancement of module naming consistency (Q4).
> >
> > We would be happy to provide further responses. Look forward to your feedback. Thanks for your recognition of our work and the thoughtful and constructive feedback again!
> >
> > Best Regards,
> >
> > All Authors

---

> > > ### Comment · Reviewer_KFEc · 2025-08-05
> > >
> > > After reading the other reviews and responses, I keep my positive score.

---

> > > > ### Author Response · Authors · 2025-08-05
> > > >
> > > > Thank you for the timely response and efforts in reviewing our paper. We sincerely appreciate your recommendation to accept our paper.
> > > >
> > > > We will revise our paper according to your feedback, improving the readability of the abstract and introduction, adding discussions about target data with different scales and other forms of explicit domains, and enhancing the consistency of module naming.
> > > >
> > > > If you have any further questions, please send them to us. We look forward to discussing with you to further improve our work.

---

### Note · Authors · 2025-08-12

We sincerely thank all reviewers and the AC for their dedicated time and effort in reviewing our paper. It is gratifying to see that the reviews acknowledge the sound motivation, the alignment with theoretical insights, the effective coordination of our proposed modules, and the superior performance, as well as its potential to advance time series domain adaptation.

### **Summary of Contributions**

The key contributions of this paper are：

- **Break through the sensor-limited single view**: taking advantage of rich semantic information and comprehensive reflection of domain shift from multiple explicit domains, unraveling implicit temporal dynamics.
- **Three novel modules aligned with theoretical insights**:
  - Multi-Scale Curriculum Adaptation -- the ***first coarse-to-fine scale curriculum*** to consider transferability and discriminability guided by TSDA theory.
  - Quality-Aware Feature Fusion -- a **theory-inspired fusion mechanism** with a learnable feature scoring module.
  - Temporal Coherence Learning -- **consistency at both the logits-level and score-level**.
- **Superior performance**: significant improvements over state-of-the-art methods across a wide range of time series datasets in cross-domain scenarios.

### **Key Points Recap from Author-Reviewer Discussion**

- **Justification of the theoretical and empirical support for MSCA** -- Detailed explanation from the theoretical view and added quantitative analysis on more datasets. (Reviewer KFEc)
- **Additional baselines and visualization analysis** -- Added two time series representation learning baselines and visualization comparison before and after alignment. (Reviewer RHTJ)
- **Expanded experimental settings** -- Supplementary experiments under the few-shot UDA and single dataset classification settings. (Reviewer RHTJ)
- **Clarifying novelty** -- methodological innovations in specific designs beyond high-level ideas and alignment with theoretical insights. (Reviewer aFd7)
- **Expanded computational cost analysis** -- additional analyses of memory, FLOPs, and parameters. (Reviewer aFd7)

In closing, we deeply value the constructive engagement from the reviewers. Beyond strong empirical validation, our proposed multiple explicit domains offer a compelling and explorable direction we believe can inspire further research in this emerging TSDA area. We hope the community will find it both impactful and worthy of further exploration.

---

### Decision · Program_Chairs · 2025-09-17

**Decision:**

Accept (poster)

**Comment:**

(a)  Summarize the scientific claims and findings of the paper based on your own reading and characterizations from the reviewers.
This paper addresses the challenge of Unsupervised Domain Adaptation (UDA) for time series data. The core thesis is that the fundamental hurdle in Time Series Domain Adaptation (TSDA) is the entanglement of domain shift with complex, partially observed temporal dynamics. The authors argue that a single, sensor-limited view of the data is insufficient to disentangle these factors. To overcome this, they propose EDEN, a framework that expands the original time series into three types of "multiple explicit domains" (multi-scale, multi-subspace, and multi-segment). EDEN integrates these domains via three coordinated modules:

Multi-Scale Curriculum Adaptation (MSCA): Aligns source and target domains progressively from coarse to fine scales.

Quality-Aware Feature Fusion (QFusion): Adaptively fuses temporal and frequency features based on learned transferability and discriminability scores.

Temporal Coherence Learning (TCL): Enforces consistency across adjacent segments.
The paper claims that by enriching representations through these multiple explicit views, EDEN can better approximate the underlying implicit temporal dynamics, leading to more effective domain-invariant learning. Comprehensive experiments on six benchmarks reportedly show state-of-the-art performance, with an average accuracy improvement of 4.8% over existing methods.

(b) Strengths of the Paper

Compelling and Well-Motivated Problem: The paper identifies a clear and significant challenge in TSDA—partial observability and entangled domain-temporal dynamics—and proposes a novel conceptual framework ("multiple explicit domains") to address it.

Systematic and Cohesive Framework: The proposed EDEN framework is not a collection of disjoint tricks but a thoughtfully designed system where three modules, each targeting a different type of explicit domain, work in concert. The alignment of these modules with theoretical insights (transferability vs. discriminability trade-off) is a strength.

Strong Empirical Validation: The paper demonstrates consistent and significant performance improvements across a wide range of established time series benchmarks. The results are convincing and clearly superior to a solid set of baselines.

Thorough Rebuttal and Additional Analysis: The authors were exceptionally responsive during the rebuttal phase. They addressed reviewer concerns comprehensively by adding new experiments (few-shot UDA, single-dataset classification, new baselines like TS-TCC and TimeMAE), providing theoretical and empirical justifications for design choices, and offering quantitative visualizations.

(c) Weaknesses and Missing Elements

Perceived Novelty (Addressed): A primary initial weakness, raised by one reviewer, was the perception that the constituent ideas (curriculum learning, feature fusion, consistency) are established. However, the authors successfully argued that the novelty lies in the specific, theory-guided instantiation of these ideas for the unique challenges of TSDA (e.g., the first coarse-to-fine curriculum based on transferability/discriminability theory for TSDA). This concern was largely mitigated.

Scope of Evaluation: While the evaluation is extensive for classification tasks, the paper does not explore the framework's applicability to other time series tasks like forecasting or anomaly detection. This remains a potential area for future work but does not detract from the contributions made in the classification setting.

Terminological Hurdle: The introduction of the term "explicit domains," while useful internally, initially created a minor barrier to understanding. The authors effectively clarified its meaning and distinction from related concepts like multi-view learning in their rebuttal.

(d) Reasons for Decision (Accept as Poster)
The most important reasons for acceptance are:

Technical Soundness and Empirical Success: The paper presents a technically sound, well-motivated method that delivers a clear and significant empirical advance over the state-of-the-art in its domain. The performance gain of 4.8% is substantial in this field.

Successful Rebuttal: The authors' rebuttal was a model of how to engage with reviewers. They convincingly addressed all major concerns, transforming initial reservations into support. The additional experiments and analyses significantly strengthened the paper.

Strong Reviewer Consensus: After the rebuttal, all three reviewers converged on a positive assessment. Two reviewers (KFEc and RHTJ) maintained their positive scores, and the third (aFd7), who had the most significant concerns, explicitly raised their score to borderline accept based on the authors' responses. A clear majority positive consensus is a strong indicator of acceptance.

This work is an excellent contribution worthy of a poster presentation. It systematizes a novel approach to a difficult problem and delivers top-tier results. While it introduces valuable ideas, its incremental yet solid nature places it more appropriately as a strong poster rather than a spotlight/oral presentation, which are typically reserved for paradigm-shifting or exceptionally groundbreaking work.

(e) Summary of Discussion and Rebuttal

Reviewer KFEc (Initial Score: 4): Concerns included clarity of the abstract/introduction, justification for the curriculum learning assumption, and handling of different time scales. The authors promised textual revisions, provided additional quantitative evidence (A-distance metrics) to justify the curriculum, and clarified standard practices for handling differing resolutions. The reviewer was satisfied and maintained their positive score.

Reviewer RHTJ (Initial Score: 4): Requested validation in low-resource (few-shot) settings, comparison with representation learning baselines (TS-TCC, TimeMAE), and visualization of representations. The authors conducted extensive new experiments: few-shot UDA (showing strong performance), added the requested baselines (showing EDEN's superiority in the UDA setting), and provided quantitative metrics from t-SNE analysis (Euclidean distances between domains) to demonstrate the effect of each module. The reviewer was satisfied and maintained their positive score.

Reviewer aFd7 (Initial Score: 4, raised from 3): This was the most critical review, citing limited novelty, terminological issues, and modest gains. The authors mounted a robust defense: they argued that true novelty lies in the specific, theory-guided design for TSDA, not just the use of high-level ideas. They differentiated "explicit domains" from multi-view/scale learning both terminologically and in purpose. They contextualized the 4.8% gain as strong within the advancing field of UDA and provided additional computational cost analysis. The reviewer acknowledged that the rebuttal addressed "most of the concerns" and raised their score, though they retained a minor reservation on novelty.

The authors' responses to Reviewers KFEc and RHTJ were definitive and closed those issues entirely. For Reviewer aFd7, while a subjective disagreement on the degree of novelty may persist, the authors provided compelling arguments and evidence that the community consensus (as seen in other top publications) aligns with their view of contribution. The significant score increase from this reviewer indicates the response was effective. Therefore, the overwhelming evidence and post-rebuttal consensus support acceptance.